# Characteristics, primary sources and secondary formation of water soluble organic aerosols in downtown Beijing

Qing Yu[1,2], Jing Chen[1,2], Weihua Qin[1,2], Siming Cheng[1,2], Yuepeng Zhang[1,2], Yuewei Sun[1,2], Ke Xin[1,2], Mushtaq Ahmad[1,2]

[1]State Key Joint Laboratory of Environment Simulation and Pollution Control, School of Environment, Beijing Normal University, Beijing 100875, China.
[2]Center of Atmospheric Environmental Studies, Beijing Normal University, Beijing 100875, China.

*Correspondence to*: Jing Chen (jingchen@bnu.edu.cn)

**Abstract.** Water soluble organic compounds (WSOC) account for a large proportion of aerosols and play a critical role in various atmospheric chemical processes. In order to investigate the primary sources and secondary production of WSOC in downtown Beijing, the day and night $PM_{2.5}$ samples in January (winter), April (spring), July (summer) and October (autumn) of 2017 were collected and analyzed for WSOC and organic tracers in this study. WSOC was dominated by its moderately hydrophilic fraction, and showed the highest concentration in January and comparable levels in April, July and October of 2017. Some typical organic tracers were chosen to evaluate the emission strength and secondary formation of WSOC. Seasonal variation of the organic tracers suggested significantly enhanced formation of anthropogenic secondary organic aerosols (SOA) during the sampling period in winter, and obviously elevated biogenic SOA formation during the sampling period in summer. These organic tracers were applied into positive matrix factorization (PMF) model to calculate the source contributions of WSOC as well as its moderately and strongly hydrophilic portions. The secondary sources contributed more than 50 % to WSOC, with higher contributions during the sampling periods in summer (75.1 %) and winter (67.4 %), and the largest contributor was aromatic SOA. Besides, source apportionment results under different pollution levels suggested that controlling biomass burning and aromatic precursors would be effective to reduce WSOC during the haze episodes in cold seasons. The impact factors for the formation of different SOA tracers, total secondary organic carbon (SOC) as well as moderately and strongly hydrophilic SOC were also investigated. The acid-catalyzed heterogeneous or aqueous-phase oxidation appeared to dominate in the SOC formation during the sampling period in winter, while the photochemical oxidation played a more critical role during the sampling period in summer. Moreover, photooxidation played a more critical role in the formation of moderately hydrophilic SOC, while the heterogeneous or aqueous-phase reactions posed more vital effects on the formation of strongly hydrophilic SOC.

# 1 Introduction

Organic compounds account for a considerable fraction (20-60 %) of atmospheric aerosols (Huang et al., 2014; Zhang et al., 2020), and water-soluble organic carbon (WSOC) generally composes 30-70 % of organic carbon (OC) (Zhang et al., 2018; Yang et al., 2019; Chen et al., 2020). WSOC in aerosols is active in light adsorption (Yan et al., 2015; Geng et al., 2020), thus may make significant impact on radiative forcing and global climate change (Andreae and Gelencser, 2006). Meanwhile, the photoexcitation of water-soluble brown carbon (BrC) can generate oxidants in aerosols and cloud/fog droplets (Manfrin et al., 2019; Kaur et al., 2019), which can promote atmospheric chemical reactions and aging processes of organic aerosols. Overall, WSOC is widely involved in the cloud processes and heterogeneous reactions due to its surface activity and water solubility (Ervens et al., 2011; George et al., 2015), thus plays a significant role in severe haze episodes (Cheng et al., 2015; Wu et al., 2019; Ma et al., 2020). Besides, WSOC is closely linked to the oxidative potential of aerosols, posing adverse health outcomes (Verma et al., 2012; Chen et al., 2019; Wang et al., 2020). Therefore, it is of great significance to study the characteristics, primary sources and secondary production of WSOC in atmospheric particulate matter.

Based on the solid phase extraction (SPE) by the Oasis HLB column, WSOC can be divided into the moderately hydrophilic fraction and strongly hydrophilic fraction (Varga et al., 2001; Kiss et al., 2002). The moderately hydrophilic fraction of WSOC mainly consists of humic-like substances (HULIS), which are an unresolved mixture of polycyclic ring structures with substituted hydroxyl, carboxyl, carbonyl, methoxy, and ester groups (Kiss et al., 2002; Lin et al., 2010; Fan et al., 2012). In addition, some smaller molecules with well-defined structures, such as phthalic acid and suberic acid, may also constitute a minor portion of the moderately hydrophilic fraction of WSOC (Lin et al., 2010). The strongly hydrophilic fraction mainly consists of low-molecular-weight organic acids (such as oxalic acid, succinic acid, malic acid) and anhydrosugars (such as levoglucosan, xylose, sucrose) (Lin et al., 2010). Previous studies have revealed that the moderately and strongly hydrophilic fractions of WSOC show significantly different intrinsic oxidative potentials, thus would pose different effects on human health (Verma et al., 2012, 2015; Yu et al., 2018). However, source contributions of the moderately and strongly hydrophilic WSOC were scarcely investigated and compared in previous research.

So far, the most widely used source apportionment approaches of organic aerosols include the chemical mass balance (CMB) model coupled with tracer-yield method (Guo et al., 2012; Islam et al., 2020), and the aerosol mass spectrometry combining positive matrix factorization (AMS-PMF) method (Hu et al., 2017; Sun et al., 2018; Shen et al., 2019). CMB model is used to quantify the primary sources of organic aerosols, while the tracer-yield method is applied to calculate the contributions of secondary sources. However, CMB model requires local source profiles, and the tracer-yield experiments conducted under simple chamber conditions usually ignore cloud and aqueous-phase processes, leading to large uncertainties when applying these results to the real atmosphere (Kleindienst et al., 2007; Feng et al., 2013). The AMS-PMF method is based on the mass spectra of organic aerosols and massive data analysis, which can avoid such disadvantages. However, online AMS can not

differentiate WSOC from water-insoluble OC, not to mention the moderately hydrophilic and strongly hydrophilic WSOC. Besides, AMS-PMF method typically classifies secondary organic aerosols (SOA) into two categories, the less oxidized oxygenated organic aerosols (LO-OOA) and the more oxidized oxygenated organic aerosols (MO-OOA), which is unable to distinguish SOA from different precursors or sources in most cases. To raise effective control measures targeting on the specific SOA precursors, some recent studies introduced the SOA tracers into PMF model to investigate the secondary sources of organic aerosols (Kang et al., 2018a, b; Geng et al., 2020).

Previous studies have suggested that coal combustion (Zhang et al., 2018; Li et al., 2019a, c), traffic emissions (Kawamura and Kaplan, 1987; Li et al., 2019c), residual oil combustion (Kuang et al., 2015), cooking (Qiu et al., 2020), soil dust and sea salts (Huang et al., 2006) can all contribute to WSOC. However, it is most commonly recognized that WSOC mainly derives from biomass burning and SOA (Ding et al., 2008; Feng et al., 2013; Du et al., 2014; Zhang et al., 2018). For example, Du et al. (2014) suggested that SOA, biomass burning and other primary combustion sources contributed about 54 %, 40 % and 6 % respectively to WSOC in Beijing during 2010-2011. Zhang et al. (2018) also indicated that the sum of biomass burning and SOA contributed more than 80 % to WSOC in Beijing, Shanghai, Guangzhou and Xi'an in the wintertime of 2013. In recent years, the adjustment of energy and industrial structures as well as the effective control of the open burning activities in the surrounding areas of Beijing have posed significant impact on the source emissions. The average $PM_{2.5}$ concentration in Beijing has been greatly reduced from 89.5 $\mu g \cdot m^{-3}$ in 2013 to 58.0 $\mu g \cdot m^{-3}$ in 2017 since the implement of the Action Plan of Air Pollution Prevention and Control in 2013 (Cheng et al., 2019). Meanwhile, it has been reported that the oxidant concentrations were enhanced accompanying the decrease of $PM_{2.5}$ level, which might promote the SOA formation (Feng et al., 2019). Consequently, the sources and composition of WSOC in Beijing may show significant changes due to the control policies and enhanced atmospheric oxidizing capacity in the surrounding areas in recent years. Therefore, it is necessary to compare the source contributions of WSOC with those in the previous studies.

A large fraction of WSOC is formed in the atmosphere, and WSOC greatly overlaps with SOA (Zhang et al., 2018). Because most of the organic aerosols remain unidentified at a molecular level, it is difficult to thoroughly understand the formation mechanisms of secondary WSOC or SOA. An effective approach to explore the formation mechanisms of SOA is to classify SOA into several categories based on their specific properties. For example, the formation mechanisms of secondary organic carbon (SOC) in the moderately and strongly hydrophilic fractions of WSOC may differ from each other due to their different water-solubility in the cloud droplets or the aqueous phase of aerosols. Furthermore, the formation mechanisms may also be different for SOA from different precursors, which originate from various sources and show disparate chemical structures and properties (Sun et al., 2016; Cheng et al., 2018). While gas-phase photooxidation of volatile organic compounds (VOCs) is an important formation pathway of SOA, direct observations have proved that the aqueous-phase reactions of VOCs from biomass burning contribute remarkably to SOA formation (Gilardoni et al., 2016). In this regard, SOA tracers can provide implications for, though may not fully represent, the formation mechanism of SOA from corresponding precursors. Besides, in recent years, haze episodes still occur frequently in Beijing in cold seasons. The humid

meteorological conditions as well as the high concentrations of VOC precursors and HONO during haze episodes may pose unique effects on the formation of some secondary components in WSOC (Li et al., 2019b; Yang et al., 2019; Zhang et al., 2019). Previous research found that haze events were usually accompanied with elevated WSOC/OC ratio (Cheng et al., 2015; Yang et al., 2019) and enhanced SOA production (Huang et al., 2014; Li et al., 2019b). Comparing source contributions of WSOC under different pollution levels would help to better understand the complex properties of WSOC, and put forth effective control measures to reduce WSOC during haze events.

In this work, the 12-hour day and night PM$_{2.5}$ samples in Beijing in January, April, July and October of 2017 were collected and analyzed for WSOC and organic tracers. The characteristics of WSOC and the selected organic tracers were investigated, and the contributions of primary and secondary sources to WSOC and its moderately and strongly hydrophilic fractions were quantified. The key influencing factors for the formation of different SOA tracers, the moderately hydrophilic SOC and strongly hydrophilic SOC were explored and compared, so as to gain insights into the possible formation mechanism of different types of SOA.

## 2. Experimental

### 2.1 Sampling

Field sampling was performed on the roof of a twenty-meter-high building in the campus of Beijing Normal University in downtown Beijing, which is considered to be a representative urban site. Fine particulate matter (PM$_{2.5}$) were sampled in four seasons of 2017 using a high-volume sampler (TH-1000C, Wuhan Tianhong Instruments Co. Ltd, China) equipped with an PM$_{2.5}$ impactor at a flow rate of 1.05 m$^3$ min$^{-1}$. The potential sampling error was discussed and estimated in the supplementary materials (S1). The sampling periods included January 2nd-16th, April 7th-23rd, July 3rd-18th and October 12th -28th in winter, spring, summer and autumn of 2017, respectively. Sampling was conducted during 8:00-19:30 in the daytime and during 20:00-7:30 at night. A total of 124 effective PM$_{2.5}$ samples were obtained in this study. The field blank samples were collected before and after each sampling period, and a total of 8 field blank samples were obtained. The blank filters were put on the filter holder of PM$_{2.5}$ sampler without pumping for 1 min, then stored and analyzed together with the ambient samples. All the samples were gathered on the quartz filters (PALLFLEX) which were pre-baked at above 500 °C for at least 4 hours before sampling. Before and after sample collections, the quartz filters were weighed by an analytical balance for three times after stabilizing under conditions of fixed temperature (20 ± 1 °C) and humidity (40 ± 2 %) for 24 h. After that, the sampled filters were stored under dark conditions below -20 °C until being analyzed.

## 2.2 Chemical analysis

To measure the values of OC and EC, a part (0.296 cm$^2$) of each filter was detected using a DRI 2001A carbon analyzer with thermal/optical reflectance (TOR) protocol. The analysis of WSOC and water-soluble ions followed the same procedure as in our previous research (Chen et al., 2014), and the details can be found in the supplementary materials (S2-1).

WSOC was further divided into its moderately and strongly hydrophilic fractions by the solid phase extraction (SPE). The moderately hydrophilic fraction of WSOC (MH-WSOC) was directly measured by the following procedure. Briefly, a punch of the sampled filter was shredded into tiny pieces and extracted using 20 mL ultrapure water for a duration of 30 min, and then filtrated via a 0.45 μm PTFE filter. The extract was acidified to pH=2.0 with HCl (1mol L$^{-1}$), then passed through a SPE cartridge (Oasis HLB, 30 μm, Waters). After that, the SPE column was rinsed with 3 mL water, then eluted with 1.5 mL

MeOH (containing 3 % NH$_3$) for three times. The eluent was blown to dryness and redissolved in 20 mL ultrapure water, and measured by a total organic carbon (TOC) analyzer (Shimadzu TOC-L CPN). The strongly hydrophilic fraction of WSOC (SH-WSOC) was calculated as the total WSOC minus the moderately hydrophilic WSOC.

Seven organic tracers, including levoglucosan, cholesterol, 4-methyl-5-nitrocatechol, phthalic acid (Ph), 2-methylerythritol, 3-hydroxyglutaric acid, and *cis*-pinonic acid (details shown in Section 3.2), were analyzed for each sample in this study. A

punch of each filter was ultrasonically extracted twice in 10 mL MeOH for 20 min (below 20 ℃). The combined extracts were filtrated, concentrated and stored in dark place at -20 °C until further derivation. Afterwards, the concentrates were blown to entire dryness with gentle ultrapure nitrogen (N$_2$), then redissolved in 100 μL pyridine and reacted with 200 μL silylating reagent (BSTFA with 1 % TMCS) at 75°C for 70 min. After cooling down to the room temperature, the derivative products were diluted with n-hexane and immediately analyzed using the GC/MS/MS equipped with a JA-5MS column. The

detailed parameter setting of GC/MS/MS can be found in the supplementary materials (S2-2). The authentic standards (Table S2) were dissolved in anhydrous pyridine, and diluted to five to seven different concentrations. Then 100 μL of the standard solutions were reacted with 200 μL silylating reagent (BSTFA: TMCS = 99:1) at 75°C for a duration of 70 min. After cooling down to the ambient temperature, these solutions containing derivative products were diluted to 1 mL with n-hexane, and measured by GC/MS/MS right before the analysis of ambient samples. The R$^2$ of the derivative products were all above

0.99, indicating good linearities of these standard curves.

To ensure quality of the measurement, the recovery rates were determined together with ambient samples, by measuring the authentic standards (Table S2) spiked onto the pre-combustioned quartz filters. The recovery rates were all in the range of 70 %-110 %, and the relative standard deviations (RSD) were all below 15 %. Besides, since two field blank samples were obtained during each sampling period, the average concentrations of the targeted compounds on these two field blanks were

used for the correction in the corresponding season. The concentrations of the targeted compounds on the field blanks were all close to zero. The reported concentrations of the targeted compounds were: the measured concentrations on each ambient sample minus the average concentration on the two field blank samples.

**2.3 PMF source apportionment**

The positive matrix factorization (PMF) model is an effective multivariate factor analysis tool which can decompose a matrix of speciated sample data (X) into two matrices: source contributions (G) and source profiles (F) (Norris et al., 2014). In this study, PMF 5.0 was applied for the source apportionment of WSOC in $PM_{2.5}$ during the sampling periods in four seasons of 2017 with a total of 124 samples. Seventeen species were input into PMF model, including WSOC, MH-WSOC, SH-WSOC, elemental carbon, sulfate, nitrate, oxalate, ammonium, magnesium, calcium and seven organic tracers. The PMF model was run repeatedly by changing the number of factors, and the derived solutions were compared. The base solution was selected based on: (1) the interpretability of the factor profiles and temporal variations of source contributions; (2) the reconstruction of the total variable and the $R^2$ of the input organic tracers ($R^2$>0.90); (3) the scaled residuals of the input species. Afterwards, the selected base solution was subjected to the displacement (DISP) and bootstrap (BS) tests for error estimation. To reduce the variability of the base solution, some constraints were defined based on the priori information on the sources (Norris et al., 2014; Paatero et al., 2014; Bozzetti et al., 2017). The detailed information about the uncertainty calculation for the input data, the selection criteria for the optimal solution, the diagnostics and the error estimates are provided in the supplementary materials (S3).

**2.4 Other data collection and calculations**

During the sampling periods, the meteorological parameters were simultaneously monitored using a HOBO meteorological station at our sampling site. The hourly concentrations of $O_3$ and CO were obtained from a nearby urban air monitoring station (3.4 km from the sampling site) via the website at http://www.bjmemc.com.cn. These data were transformed to 12-h averages corresponding to the sampling time.

The liquid water content (LWC) in inorganic aerosols was calculated by the ISORROPIA-II model (Fountoukis and Nenes, 2007), and the reverse mode was chosen in this study since the concentrations of gaseous pollutants such as HCl, $HNO_3$ and $NH_3$ were not available here. The total aerosol LWC was the summation of the water in both water-soluble ions and organic species, and the latter was calculated by the approach described by Cheng et al. (2016). To estimate the aerosol acidity, the approximate value of aerosol pH ($pH_F$) was also estimated by the ISORROPIA-II model (Pye et al., 2020). The molality of $H^+$ ($m_{H^+}$), which was calculated by: $m_{H^+} = 10^{-pH_F}$, was used for the correlation analysis.

The OC emission amounts from open biomass burning over the sampling periods in the four seasons of 2017 were obtained from the Fire Inventory (FINNv1.5), which provides daily estimates of the OC emissions from wildfire and agricultural fires with a resolution of 1 km (Wiedinmyer et al., 2011). These emission data were processed by the fire_emis utility provided on http://bai.acom.ucar.edu/Data/fire/. The values of OC emissions in the Beijing-Tianjin-Hebei region were extracted using the Geographic Information System (GIS).

## 3 Results and discussion

### 3.1 Temporal trends of carbonaceous species

The temporal variations of WSOC, OC, EC and PM$_{2.5}$ during the sampling periods in the four seasons of 2017 are shown in Figure 1, and their average concentrations are summarized in Table 1. In general, the carbonaceous species showed similar variation trends with that of PM$_{2.5}$ throughout the whole sampling period, which approximately varied in the periodic cycles of two to seven days (Guo et al., 2014). The average values of WSOC, OC, EC and PM$_{2.5}$ in January were all much higher than those during the sampling periods in other seasons. The mean levels of OC, EC and PM$_{2.5}$ were in the descending order

of October > April > July, while WSOC showed comparable levels over the sampling periods in these three seasons. The fact that WSOC exhibited mild temporal variation with no sudden increase in October implied that short-term outdoor biomass burning after the harvest season in the surrounding areas of Beijing was well controlled over the sampling period in autumn. As shown in Table S5, the OC concentration in PM$_{2.5}$ in Beijing exhibited an overall declining trend in the past ten years, but the WSOC level showed no obvious change. Therefore, the control of WSOC seemed to be more challenging compared to

the control of water-insoluble organic aerosols.

The temporal variation of the WSOC/OC ratio is also illustrated in Figure 1. WSOC/OC exhibited the highest average value during the sampling period in summer, followed by that in winter. The range of WSOC/OC in this study was similar to that previously reported in urban Beijing (Zhao et al., 2018; Yang et al., 2019). Previous studies showed that the aggravation of PM$_{2.5}$ pollution was usually accompanied by the elevated WSOC/OC ratio (Cheng et al., 2015; Yang et al., 2019), which was

200 also confirmed by the significant positive correlations between PM$_{2.5}$ and the WSOC/OC ratio during the sampling periods in four seasons (winter: r=0.68, p<0.01; spring: r=0.61, p<0.01; summer: r=0.36, p<0.05; autumn: r=0.91, p<0.01) in this study, again underling the important role of WSOC during the haze evolution process. In this work, WSOC was further divided into its moderately and strongly hydrophilic fractions. As listed in Table 1, the moderately hydrophilic WSOC dominated in the total WSOC throughout the sampling period, and showed the highest proportion in July (0.84), followed by January (0.73),

higher than the results previously reported in Beijing (Li et al., 2019a; Huang et al., 2020). Besides, the ratio of the strongly hydrophilic WSOC to the total WSOC showed significant positive correlations with PM$_{2.5}$ in January (r=0.58, p<0.01), July (r=0.48, p<0.05) and October (r=0.44, p<0.05).

### 3.2 Characteristics of organic tracers

#### 3.2.1 Seasonal variations and diurnal patterns

The average concentrations of the organic tracers identified in this study are summarized in Table 1. Levoglucosan is a tracer for biomass burning, while cholesterol is a good indicator of cooking, and both of them are known as primary organic tracers. Previous studies have shown that phthalic acid can be used as an aromatic SOA tracer (Al-Naiema and Stone, 2017; Huang et al., 2019), while 4-methyl-5-nitrocatechol can be used as a tracer for biomass burning SOA (Iinuma et al., 2010; Bertrand

et al., 2018; Srivastava et al., 2018), and both of them serve as the anthropogenic SOA tracers. The biogenic SOA tracers in this study included the isoprene SOA tracer 2-methylerythritol, as well as *cis*-pinonic acid and 3-hydroxyglutaric acid, which are lower- and higher-generation oxidative products of monoterpenes respectively (Kourtchev et al., 2009). To correct for the different atmospheric diffusion conditions and better discuss the differences in emission strengths or secondary production rates in four seasons, the concentrations of the identified organic tracers were divided by the level of carbon monoxide (CO), which has relatively constant emission rate and is inert to chemical reactions. The CO-scaled concentration of these organic tracers during the sampling periods in four seasons are presented in Figure 2 (a), and the day to night ratios of the measured concentrations of these organic tracers (not normalized by CO) are shown in Figure 2 (b).

**Primary organic tracers.** The average CO-scaled concentration of levoglucosan in July was much lower than those during the study periods in other three seasons (only 15.4 % of that in January), suggesting that biomass burning might be not active during the study period in summer, similar to the findings in previous studies (Sun et al., 2018; Duan et al., 2020). The result of the variance analysis (ANOVA) indicated that levoglucosan exhibited similar CO-scaled concentrations in January, April and October ($p > 0.05$). As shown in Figure 2 (c), the OC emissions from open biomass burning in the Beijing-Tianjin-Hebei region (provided by the FINN inventory) were 302 Mg (January), 1557 Mg (April), 1818 Mg (July) and 501 Mg (October), respectively, completely different from the seasonal variation trend of levoglucosan, suggesting that open biomass burning was not the major type of biomass burning in Beijing. Previous studies indicated that residential biofuel combustion might be the main source of levoglucosan in Beijing (Chen et al., 2017), and residential coal burning might also contribute a minor fraction of levoglucosan in winter (Yan et al., 2018). The CO-scaled concentration of cholesterol during the study periods in spring and autumn were significantly higher (ANOVA, $p < 0.05$) than that in winter and summer. Some previous research found that both the regular cooking emissions and open barbecues might influence the organic aerosols in Beijing (Xu et al., 2015). The enhanced CO-scaled concentrations of cholesterol in spring and autumn were perhaps due to the enhanced open barbecues in spring and autumn when the weather was mild.

**Anthropogenic SOA tracers.** Phthalic acid showed the highest CO-scaled concentration in January, followed by July. The highest phthalic acid over the study period in winter might be due to stronger aromatic SOA formation. In the previous study, a significant increase of toluene-derived SOA in winter was universally observed in the northern cities of China (Ding et al., 2017). In winter, the enhanced emissions due to residential heating and the adverse atmospheric diffusion condition led to higher concentrations of PAHs (Feng et al., 2018; Sun et al., 2018), thus might facilitate the PAH-derived SOA formation. Although the concentrations of oxidants were usually lower in winter due to the weak solar radiation, a previous observation found that the ·OH concentration in Beijing was significantly higher than that in New York, Birmingham and Tokyo, and was nearly 1 order of magnitude larger than that predicted by global models in northern China in winter (Tan et al., 2018). Zhang et al. (2019) indicated that HONO, which was mainly from the heterogeneous reactions of $NO_2$ and traffic emissions, was the major precursor of OH radicals in winter. According to the WRF-Chem simulation, HONO resulted in a significant enhancement (5-25 $\mu g \ m^{-3}$) of SOA formation (most of which were from the aromatic precursors) during a haze episode in

winter in the Beijing-Tianjin-Hebei region (Zhang et al., 2019). Besides, some recent studies suggested that brown carbon-derived singlet molecular oxygen ($^1O_2^*$) in aerosol liquid water could react rapidly with the electron-rich organics such as PAHs, thus facilitate the formation of aromatic SOA (Kaur et al., 2019; Manfrin et al., 2019). This process might be more significant during the sampling period in winter, when the HULIS concentration was much higher than that in other seasons. Furthermore, the high aerosol LWC during the sampling period in winter can also promote the heterogeneous or aqueous-phase formation of aromatic SOA. Except for stronger aromatic SOA formation, the enhanced gas-to-particle partition of phthalic acid due to low temperature, high relative humidity and high OA loadings in winter might also be one of the reasons for the highest phthalic acid in January (Huang et al., 2019). The second highest value of phthalic acid occurring in July was probably due to high temperature and relative humidity, strong solar radiation and abundant oxidants in summer, which was favorable for the secondary photochemical reactions (Zhao et al., 2018). As presented in Figure 2 (b), the day to night ratio of phthalic acid was obviously higher than 1 in April, July and October. Since the atmospheric diffusion condition was worse at night (Table S6), higher concentrations of phthalic acid in the daytime could be attributed to stronger secondary formation (Kawamura and Yasui, 2005). The day/night ratio of phthalic acid in July was much higher than those in other study periods, which was possibly due to more prominent effect of photochemical processes on the generation of aromatic SOA in summer (Kawamura and Yasui, 2005).

4-Methyl-5-nitrocatechol was usually below detection limit for the summer samples, again indicating that biomass burning was not active in July. It is notable that the CO-scaled concentrations of 4-methyl-5-nitrocatechol in January were much higher than those in April and October, similar to the result of Kahnt et al. (2013), while the primary emission intensity of biomass burning was relatively constant during the sampling periods in these seasons as revealed by the seasonal distribution pattern of levoglucosan. This phenomenon can be explained by the following reasons. First of all, the adverse atmospheric diffusion condition in winter favored the accumulation of its precursors such as phenols and $NO_x$ (Iinuma et al., 2010), thus might increase the formation rate of biomass burning SOA. Secondly, the low temperature and high aerosol LWC in winter were beneficial for the partitioning of the gaseous phenols to the aerosol phase. Besides, the high concentration of HONO in winter might also contribute to the formation of nitrocatechols in the condensed phase (Vidovic et al., 2018; Qu et al., 2019). Hence, the aqueous-phase or heterogeneous reactions were enhanced in winter (Li et al., 2014; Gilardon et al., 2016).

**Biogenic SOA tracers.** As presented in Figure 2, 2-methylerythritol and 3-hydroxyglutaric acid showed extremely high CO-scaled concentrations in July, followed by April and October, and showed the minimum concentrations in January. The mean concentration of 2-methylerythritol during the study period in summer was almost 50 times of that in winter. Such variations of biogenic SOA tracers indicated much stronger biogenic SOA formation during the sampling period in summer, which was attributable to the higher emissions of biogenic precursors and accelerated photochemical reactions (Shen et al., 2015; Qiu et al., 2020). The day/night ratios of 2-methylerythritol and 3-hydroxyglutaric acid were higher than 1 in July, and significantly higher than those in other seasons, indicating that photochemical oxidation played a more significant role in their formation processes during the sampling period in summer. Nevertheless, *cis*-pinonic acid, another monoterpene SOA tracer, showed

slightly higher CO-scaled concentration in April than in July (ANOVA, p<0.05). This might be due to the active atmospheric oxidation processes in summer facilitated the transformation of *cis*-pinonic acid to its higher-generation oxidation products. Besides, *cis*-pinonic acid tends to evaporate into the gas phase in summer under high temperature (Li et al., 2013; Ding et al., 2016). Different from the other SOA tracers which showed the highest day/night ratios in July, *cis*-pinonic acid exhibited the lowest day/night ratio during the study period in summer. This might be also because *cis*-pinonic acid was rapidly
transformed to its higher-generation oxidation products when the photochemical oxidation was the strongest in the daytime in summer.

**3.2.2 Influencing factors for the formation of different SOA tracers**

Spearman correlation analysis was conducted between the SOA tracers and meteorological parameters, $O_3$, aerosol acidity and aerosol LWC, and the results are shown in Table S7 and Figure 3. In fact, sometimes all the $PM_{2.5}$ components, even the
290 primary components, can correlate well with RH and LWC (Sun et al., 2013). To solve this problem, the concentration of SOA was normalized by HOA (Sun et al., 2013), CO (Kleinman et al., 2008; Aiken et al., 2009), EC (Zheng et al., 2015) or OA (Xu et al., 2017) in previous studies, to better evaluate their secondary generation. The correlation coefficients between the ratio of SOA tracers to OC and RH as well as LWC are also listed in the supplementary materials (Table S8).

The ratio of WSOC/OC showed significant positive correlations with RH, LWC and the aerosol acidity in January, April and
295 October, possibly because the acid-catalyzed heterogeneous or aqueous-phase reactions facilitated the formation of SOC in WSOC during the sampling periods in these seasons (Du et al., 2014; Yang et al., 2019). Besides, the increased aerosol LWC could facilitate the the partitioning of the gas-phase WSOC to the aerosol phase (Hennigan et al., 2009), resulting in a higher WSOC/OC ratio. No correlations between WSOC/OC and RH or LWC were found in July, likely due to enhanced gas-phase photochemical oxidation in the formation SOC in WSOC during the study period in summer. In January and October with
300 lower temperature, WSOC/OC exhibited significant positive correlations with temperature (January: r=0.50, p<0.01; October: r=0.47, p<0.01), while in April and July when the weather became warmer, no significant correlations were found between them. Higher temperature in warm seasons might inhibit the gas-to-particle partition of semi-volatile or intermediate WSOC and related heterogeneous or aqueous-phase reactions (Qian et al., 2019; Lu et al., 2019), thus counteracting its promoting effect on the formation of SOC in WSOC.

The biomass burning SOA tracer, 4-methyl-5-nitrocatechol, showed strong positive relationships with RH, LWC and aerosol acidity, and strong negative relationships with the $O_3$ concentration and solar radiation in January and April. The correlations with the above parameters became weaker in October and no data was available in July because 4-methyl-5-nitrocatechol was below the detection limit. The ratio of 4-methyl-5-nitrocatechol to OC also showed strong correlations with RH (January: r=0.73, p<0.01; April: r=0.86, p<0.01) and LWC (January: r=0.82, p<0.01; April: r=0.75, p<0.01) during the study periods
in winter and spring. Laboratory studies have revealed that phenolic compounds, which are massively emitted from biomass burning, could undergo rapid aqueous-phase oxidation and produce substantial amounts of SOA under either the simulated

sunlight (Sun et al., 2010; Li et al., 2014; Yu et al., 2014) or the dark conditions (Hartikainen et al., 2018; Kristijan et al., 2018). Direct observations have also proved that aqueous-phase reactions of the precursors from biomass burning emissions contribute significantly to SOA formation (Gilardoni et al., 2016). Therefore, the correlation results above together with the literature findings suggested that the acid-catalyzed heterogeneous or aqueous-phase reactions might play a dominant role in the formation of 4-methyl-5-nitrocatechol over the sampling periods in winter and spring. Similar speculation was also made in the previous observation (Wang et al., 2019). $O_3$ can be used as a tracer to reflect the strength of photochemical oxidation (Herndon et al., 2008; Xu et al., 2017), and the significant negative correlations between 4-methyl-5-nitrocatechol and $O_3$ in all seasons implied that the photochemical reactions might not be the major formation pathway of 4-methyl-5-nitrocatechol. Instead, stronger solar radiation might result in the photolysis of 4-methyl-5-nitrocatechol (Wang et al., 2019). Besides, the reverse relationship with temperature during the whole sampling period and in April and October was probably because that the increasing temperature was conducive to the evaporation of phenolic species, thus inhibited the secondary production of 4-methyl-5-nitrocatechol through heterogeneous or aqueous-phase reactions. The study of Gilardoni et al. (2016) also indicated that lower temperature and higher RH were conducive to the formation of biomass burning SOA.

The aromatic SOA tracer, phthalic acid, showed different correlation patterns over the sampling periods in different seasons. Chamber studies have revealed that the enhanced aerosol LWC could significantly increase the yields of aromatic SOA (Jia and Xu, 2018; Lu et al., 2019; Zhou et al., 2019). In this study, both phthalic acid and the ratio of phthalic acid/OC showed significant positive relationships with RH and LWC in January and April, suggesting that the acid-catalyzed heterogeneous or aqueous-phase reactions significantly contributed to the formation of phthalic acid during the study periods in winter and spring. In contrast, phthalic acid showed significant positive relationship with $O_3$ (r=0.65, p<0.01) and negative correlation (r=0.60, p<0.01) with RH in July, which might be due to the enhanced photochemical processes on the formation of phthalic acid during the study period in summer (Kawamura and Yasui, 2005). Different from 4-methyl-5-nitrocatechol, phthalic acid showed significantly positive correlations with temperature except in April, likely due to the higher temperature accelerated the production rates of phthalic acid (Kawamura and Yasui, 2005). Though the lower temperature might facilitate the gas-to-particle partition of phthalic acid, this process seemed not to play a dominant role during the study periods in four seasons.

The isoprene SOA tracer, 2-methylerythritol, could be formed through the gas-phase reaction with ·OH radical (Claeys et al., 2004a) and acid-catalyzed heterogeneous oxidation with $H_2O_2$ (Claeys et al., 2004b). In addition, it can also be formed by reactive uptake of the isoprene-derived epoxydiols (IEPOX) generated in the gaseous phase and subsequent aqueous-phase processing (Surratt et al., 2010; Xu et al., 2015). As shown in Table S7, during the whole study period, 2-methylerythritol exhibited significant positive relationships with temperature (r=0.60, p<0.01), RH (r=0.55, p<0.01), $O_3$ (r=0.33, p<0.01), the aerosol acidity (r=0.79, p<0.01) and LWC (r=0.56, p<0.01), suggesting that both the gas-phase photooxidation and aqueous-phase reactions played significant roles in the formation of 2-methylerythritol. From Figure 3 and Table S7, 8, it seems that the heterogeneous or aqueous-phase were more closely associated with the formation of 2-methylerythritol during the study periods in autumn and winter, while the photochemical oxidation played an enhancing role over the study period in summer.

2-Methylerythritol did not correlate significantly ($p > 0.05$) with temperature in January, April and October, however, when the temperature was above 25℃, it grew rapidly as temperature increased, similar to the result found in the previous research (Liang et al., 2012).

The monoterpene SOA tracer, 3-hydroxyglutaric acid also showed significant positive relationships with temperature ($r=0.63$, $p<0.01$), RH ($r=0.53$, $p<0.01$), $O_3$ ($r=0.37$, $p<0.01$), aerosol acidity ($r=0.82$, $p<0.01$) and LWC ($r=0.58$, $p<0.01$) during the
350 whole study period (Table S7). Besides, 3-hydroxyglutaric acid correlated strongly with 2-methylerythritol ($r=0.94$, $p<0.01$), implying similar influencing factors for the formation of these two biogenic SOA tracers. The correlation pattern of another monoterpene SOA tracer, cis-pinonic acid, was different from that of 3-hydroxyglutaric acid. Chamber studies showed that cis-pinonic acid could be produced through gas-phase reactions of monoterpenes (Yu et al., 1999; Larsen et al., 2001), which significantly contributed to the newly nucleated particles (Zhang et al., 2012). Previous field observations also found that cis-
355 pinonic acid was closely associated with the nucleation processes as the first step in the SOA formation from organic vapors (Alier et al. 2013; van Drooge et al., 2018). On the whole year scale, both cis-pinonic acid ($r=0.35$, $p<0.01$) and the ratio of cis-pinonic acid/OC ($r=0.64$, $p<0.01$) indeed exhibited significant correlations with $O_3$. However, no significant correlation was found between cis-pinonic acid and $O_3$ in July, possibly because it is an unstable intermediate and might more easily further generate higher-generation products such as 3-hydroxyglutaric acid (Kourtchev et al., 2009). And this speculation
was supported by the significant positive correlation between the ratio of 3-hydroxyglutaric acid to cis-pinonic acid and $O_3$ in July ($r=0.52$, $p<0.01$). Besides, cis-pinonic acid showed significant positive correlations with temperature in January ($r=0.37$, $p<0.05$) and October ($r=0.37$, $p<0.05$), while the correlation became weaker in April and even negative in July. At lower temperature, increasing temperature might facilitate the emission of monoterpenes and the formation rate of cis-pinonic acid. However, when the weather became warmer, the enhanced temperature might facilitate the transformation of
cis-pinonic acid to higher-generation products or its evaporation into the gas phase (Li et al., 2013; Ding et al., 2016).

### 3.3 Primary sources and secondary generation of WSOC

### 3.3.1 Source apportionment of WSOC

Source apportionment of PMF was conducted to investigate the source contributions of WSOC as well as its moderately and strongly hydrophilic fractions. Nine types of sources were identified in this study as shown in Figure 4. Factor 1 showed high
levels of levoglucosan and EC, thus was interpreted as the direct emissions from biomass burning. Factor 2 had a high level of cholesterol, hence was identified as cooking. Factor 3 exhibited a large fraction of EC which can not be explained by the direct emissions of biomass burning, suggesting that it was the direct emissions from other combustion sources, such as coal combustion, traffic and waste burning, etc. Factor 4 was characterized by high loadings of $Mg^{2+}$ and $Ca^{2+}$, thus was regarded as dust. No significant EC but high proportions of 4-methyl-5-nitrocatechol and phthalic acid were observed in Factor 5 and
Factor 6, respectively, which were identified as SOC from biomass burning (biomass burning SOC) and aromatic precursors (aromatic SOC), respectively. Factor 7 exhibited a high level of cis-pinonic acid, thus was explained as the freshly generated

biogenic SOC. Factor 8 was featured by high fractions of 2-methylerythritol and 3-hydroxyglutaric acid, which are the end oxidation products from isoprene and monoterpenes respectively, hence was recognized to be the aged biogenic SOC. Note that 3-hydroxyglutaric acid and *cis*-pinonic acid were not grouped in one factor though they are both tracers of monoterpenes, due to their different oxidation degree as discussed above. Factor 9 covered the secondary components (such as $SO_4^{2-}$, $NO_3^-$, $NH_4^+$ and $C_2O_4^{2-}$) that can not be well explained by the identified sources above, thus was considered to be SOA from other sources. More detailed discussion can be found in the supplementary materials (S3).

Source contributions to the total WSOC as well as its moderately and strongly hydrophilic fractions are illustrated in Figure 5. During the whole sampling period, the primary emissions of biomass burning contributed 23.0 % to the total WSOC, with a higher contribution to its strongly hydrophilic fraction (37.9 %) than to the moderately hydrophilic portion (15.2 %), which was probably due to the large amounts of saccharides with high water-solubility from biomass burning (Yan et al., 2019; Xu et al., 2020). The total contribution of primary and secondary biomass burning to WSOC (35.1 %) was slightly lower than those previously reported in Beijing (Cheng et al., 2013; Li et al., 2018; Duan et al., 2020), likely due to the effective control of the open biomass burning activities in the surrounding areas of Beijing in recent years. Other primary combustion sources (Factor 3) also contributed significantly to WSOC (13.0 %), most of which contributed to its moderately hydrophilic fraction (16.5 %). A recent source apportionment based on CMAQ model in North China reported that coal combustion contributed 15.1 % to water-soluble HULIS (HULISws) annually, which is the major component of moderately hydrophilic WSOC (Li et al., 2019a). High concentrations of HULISws were also observed in the coal combustion smoke, again suggesting that coal combustion might be a significant source of HULISws (Fan et al., 2016). The primary emission strength of coal combustion was the strongest in winter among four seasons, since the domestic heating required extra amounts of coal combustion in this season. However, the contribution of Factor 3 to WSOC during the study period in winter was not the highest among four seasons, implying that there could be other sources beyond coal combustion included in Factor 3. Previous studies using the PMF model also found that traffic and waste burning both contributed more than 15 % to HULISws in Beijing (Ma et al., 2018; Li et al., 2019c). Therefore, the mixed primary sources in Factor 3 possibly consisted of coal combustion, traffic emission, waste incineration, etc. Previous AMS studies in Beijing indicated that cooking contributed more than 10 % to total organic aerosols (Hu et al., 2016; Sun et al., 2018; Duan et al., 2020). However, the contribution of cooking to WSOC was quite low (2.5 %) in this study, probably because it contributed more significantly to the water-insoluble fraction of organic aerosols (Zhao et al., 2007). Our results also showed that cooking only contributed to the moderately hydrophilic WSOC. Dust contributed 7.2 % to WSOC, with 4.6 % to its moderately hydrophilic fraction and 12.1 % to strongly hydrophilic fraction. The different contributions of dust between the moderately and strongly hydrophilic fractions might be explained by the strongly hydrophilic fulvic acids from soil resuspension and the strongly hydrophilic saccharides carried by dust (Li et al., 2018).

As shown in Figure 5 (a), the secondary sources contributed 54.4 % to total WSOC, slightly higher than the results of Cheng et al. (2013) and Tao et al. (2016), and similar to that of Du et al. (2014). Secondary sources showed a higher contribution to

the moderately hydrophilic WSOC (58.5 %) than the strongly hydrophilic WSOC (44.6 %). The aromatic SOC was the most abundant secondary source of WSOC (28.8 %) as well as its moderately (25.9 %) and strongly hydrophilic fractions (33.5 %) in urban Beijing (Tang et al., 2018). The biogenic SOC (11.0 %) and biomass burning SOC (12.1 %) contributed comparable proportions to total WSOC, and both contributed more to the moderately hydrophilic WSOC. Biomass burning SOC, which is mainly generated from the phenolic compounds, has been widely recognized to contribute notably to water-soluble brown carbon (HULIS$_{WS}$) (Smith et al., 2016; Pang et al., 2020). Biogenic SOC in the strongly hydrophilic WSOC was more aged than that in moderately hydrophilic WSOC, possibly because the atmospheric aging processes would generally increase the polarity of organic aerosols (Baduel et al., 2011; Wu et al., 2016; Kuang et al., 2020).

### 3.3.2 Temporal variation of the source contributions

The seasonal variation of the source contributions to total WSOC and its moderately and strongly hydrophilic fractions are shown in Figure 5 (b). WSOC were mainly from secondary sources during the study periods in summer (75.1 %) and winter (67.4 %), but dominated by primary emissions over the sampling periods in spring (65.2 %) and autumn (71.5 %), similar to the result of Qiu et al. (2020). During the sampling period in winter, WSOC mainly originated from anthropogenic sources (Li et al., 2018; Zhang et al., 2018), including aromatic SOC (38.3 %), biomass burning SOC (27.6 %), and primary biomass burning (21.6 %). The significant contributions of anthropogenic SOC during the study period in winter probably resulted from the stronger emissions of aromatic precursors and phenolic compounds from the domestic heating activities such as the household combustion of solid fuels (Liu et al., 2016; Ding et al., 2017). The meteorological conditions over the study period in winter (slow wind speed, low temperature and high relative humidity) also favored the accumulation and heterogeneous uptake of the SOC precursors, leading to stronger SOC production. It is therefore crucial to control biomass burning and the aromatic precursors to reduce WSOC in winter. During the sampling period in summer, the largest contributor to WSOC was biogenic SOC (39.8 %). A significant fraction of the biogenic SOC was highly oxidized during the study period in summer, in consistent with the higher O/C ratio found in summer in the previous research (Hu et al., 2016; Xu et al., 2017; Qiu et al., 2020). Aromatic SOC (31.3 %) also contributed significantly to WSOC during the sampling period in summer, as reported previously in the summer of Beijing (Guo et al., 2012). Unlike the case in winter with stronger anthropogenic emissions and stagnant and humid meteorological conditions, the high fraction of aromatic SOC in summer was largely associated with the stronger photooxidation capacity. Both primary and secondary contributions from biomass burning were negligible during the study period in summer. The direct emissions of biomass burning contributed higher portions to WSOC during the study periods in spring (27.0 %) and autumn (41.8 %), with much greater contributions to the strongly hydrophilic WSOC than the moderately hydrophilic WSOC in both seasons. Besides, during the sampling period in spring when the weather was windy and dusty, dust showed the highest contribution (23.3 %) to WSOC among four seasons. The contribution of fresh biogenic SOC (13.0 %) was significantly enhanced during the sampling period in spring compared to that in winter as the temperature gradually increased, while the contribution of aromatic SOC (19.3 %) decreased dramatically compared to that in winter due to the cessation of heating activities (Yu et al., 2019).

As shown in Figure 1 as well as in previous studies, winter and autumn in North China Plain are the two seasons that endure more severe air pollution and show higher concentrations of $PM_{2.5}$. Therefore, the source contributions to WSOC during the sampling periods in winter and autumn under different pollution levels were also investigated, as shown in Figure 6. During the study period in winter, both primary and secondary contributions from biomass burning increased significantly from the clean days to moderate hazy days, while aromatic SOC and biomass burning SOC dominated during the severe hazy days (Zhang et al., 2018), highlighting the important roles of biomass burning and aromatic SOC over the hazy periods in winter (Elser et al., 2016; Li et al., 2017a; Huang et al., 2019; Yu et al., 2019). During the study period in autumn, the contribution of aromatic SOC gradually increased as the haze conditions aggravated, while the fresh biogenic SOC became less important with the aggravation of haze conditions, which might be due to the inhibited gas-phase photochemical oxidation as discussed above. Again, the change of the source contributions during the study periods in winter and autumn under different pollution levels suggested that the control of biomass burning and reduction of the aromatic precursors would be of great significance for controlling WSOC in hazy days.

### 3.3.3 Implications for the formation of moderately and strongly hydrophilic SOC

As secondary sources contributed significantly to both the moderately hydrophilic WSOC (58.5 %) and strongly hydrophilic WSOC (44.6 %), the key influencing factors for the formation of SOC in moderately and strongly hydrophilic WSOC were explored and compared. Figure 7 (a) shows the RH versus $O_3$ dependence of the ratios of moderately hydrophilic SOC to OC (MH-SOC/OC), strongly hydrophilic SOC to OC (SH-SOC/OC) and strongly hydrophilic SOC to moderately hydrophilic SOC (SH-SOC/MH-SOC) over the whole sampling period. Both the ratios of MH-SOC/OC and SH-SOC/OC increased with $O_3$ and RH, suggesting that both gas-phase photooxidation and heterogeneous or aqueous-phase reactions might play critical roles in the formation of moderately hydrophilic and strongly hydrophilic SOC. Compared to strongly hydrophilic SOC, the ratio of MH-SOC/OC was more dependent on $O_3$ (r=0.35, p<0.01), and showed slightly higher values in the daytime (paired t test, p<0.05), implying a more significant role of the photooxidation in the generation of moderately hydrophilic SOC. The laboratory studies showed that the photo-induced auto-oxidation of PAHs could lead to the formation of HULISws that can not be formed under dark conditions (Haynes et al., 2019). Besides, the photo-induced oligomerization of several phenolic compounds could also form HULISws (Vione et al., 2019). Field observations also reported an obviously higher contribution of the ultrafine aerosol mode to the moderately hydrophilic WSOC in summer, indicating that the formation of moderately hydrophilic SOC was tightly associated with the gaseous phase nucleation (Frka et al., 2018). In comparison, SH-SOC/OC ratio was more sensitive to RH (r=0.37, p<0.01) and LWC (r=0.46, p<0.01) than to $O_3$ (p>0.05), implying a more significant impact of the heterogeneous or aqueous-phase processing on the formation of strongly hydrophilic SOC. The SH-SOC/MH-SOC ratio exhibited strong correlations with RH (r=0.61, p<0.01) and LWC (r=0.60, p<0.01), which also indicated that the heterogeneous or aqueous-phase oxidation might facilitate the transformation of moderately hydrophilic SOC to the strongly hydrophilic SOC. The quantum chemical calculations in a recent study suggested that the carbenium ion-mediated reactions which involve highly hydrophilic organic species occurred efficiently in the weakly acidic aerosols and cloud/fog droplets,

contributing significantly to the SOC generation (Ji et al., 2020). The hygroscopicity of organic aerosols increased with their aging degree during their evolution processes (Jimenez et al., 2009; Wu et al., 2016; Kuang et al., 2020), therefore, the ratio of SH-SOC/MH-SOC was also employed to indicate the aging degree of SOC. The significant positive correlations between the ratio of SH-SOC/MH-SOC and RH, similar to a previous AMS study in Beijing that observed higher O/C ratios with higher RH (Xu et al., 2017), indicated that aqueous-phase oxidation might play a major role in the aging processes of SOA. Research on the aging processes in fog droplets and aerosols also supported the finding that the aqueous-phase reactions might facilitate the formation of more oxidized and strongly hydrophilic SOA (Brege et al., 2018). Previous observation suggested that a large fraction of ambient SOA was more oxidized than those formed in the dry smog chambers, in while SOA could only be produced through gas-phase oxidation (Aiken et al., 2008). There have been some hypotheses for the difference between the chamber SOA and ambient SOA, such as the losses of vapors to the walls and the autoxidation in the chamber (McVay et al., 2016; Thornto et al., 2020). The results of this study also indicated that the aqueous-phase processing, which can produce more hydrophilic SOA, may be one of the reasons for the discrepancy in the oxidation degrees of ambient SOA and chamber SOA (Ervens et al., 2011).

To further investigate the key influencing factors for the formation of moderately and strongly hydrophilic SOC during the sampling periods in four seasons, the correlation coefficients between the above ratios and some meteorological parameters, $O_3$, $PM_{2.5}$, aerosol acidity and aerosol LWC in fours seasons are illustrated in Figure 7 (b). In January, both the MH-SOC/OC and SH-SOC/OC ratios exhibited strong positive relationships with RH, aerosol acidity and LWC, suggesting that the acid-catalyzed heterogeneous or aqueous-phase reactions might be the dominant formation mechanisms of both moderately and strongly hydrophilic SOC during the sampling period in winter. Such result was consistent with the previous finding that the heterogeneous or aqueous-phase oxidation dominated in winter when the gaseous photochemical oxidation was usually weak (Duan et al., 2016; Wu et al., 2019; Yu et al., 2019). However, in July, SOC/OC exhibited significant positive correlations with $O_3$ (r=0.44, p<0.05) and temperature (r=0.54, p<0.01), implying that the photooxidation processing might be the major formation pathway of SOC in WSOC, possibly due to the stronger solar radiation in summer (Tang et al., 2016; Duan et al., 2020). However, heterogeneous or aqueous-phase oxidation might still play a critical role in transforming less oxidized MH-SOC to more oxidized SH-SOC based on the significant correlation between the SH-SOC/MH-SOC ratio and LWC (r=0.60, p<0.01). During the sampling periods in spring and autumn, similarly based on the correlation pattern of the respective ratio with the influencing factors, the production of moderately hydrophilic SOC showed a stronger linkage to the photooxidation process, while the aqueous-phase reactions might play a more critical role in the formation of strongly hydrophilic SOC and the aging processes of SOA. Besides, the SOC/OC ratio showed strong positive relationships with $PM_{2.5}$ in January (r=0.81 p<0.01), reflecting enhanced SOC formation as the pollution aggravated during the sampling period in winter (Zhang et al., 2014; Li et al., 2019b). However, no such relationship was found in July (r=0.02, p>0.05) when the gaseous photooxidation dominated in SOC formation.

**4 Conclusions**

Based on the WSOC and related SOA tracer analysis for the PM$_{2.5}$ samples collected in downtown Beijing in four seasons of 2017, the moderately hydrophilic fraction of WSOC dominated in WSOC throughout the sampling period, which showed the highest proportion to WSOC during the sampling period in summer. However, the ratio of WSOC/OC increased as pollution aggravated and the ratio of strongly hydrophilic WSOC to total WSOC increased with PM$_{2.5}$ during the sampling periods in seasons other than summer. Compared to the previous studies in Beijing over the past decades, the reduction of WSOC in this study was not as obvious as that of OC, indicating that the control of WSOC is a more challenging task than the control of water-insoluble organics.

The secondary sources contributed more than 50 % to WSOC, with higher contributions during the study periods in summer (75.1 %) and winter (67.4 %) than in spring (34.8 %) and autumn (28.5 %). Aromatic SOC (28.8 %) was the most abundant secondary source over the entire study period. Biomass burning SOC played a significant role (27.6 %) during the sampling period in winter, while biogenic SOC showed the highest contribution (39.8 %) during the sampling period in summer. The direct emissions from biomass burning and other primary combustion sources were the major primary sources of moderately hydrophilic WSOC, while strongly hydrophilic WSOC was largely affected by the direct emissions of biomass burning and dust. The total contribution of primary and secondary biomass burning to WSOC was slightly lower than those previously reported in Beijing due to the effective control of open burning in the surrounding areas. The contributions of aromatic SOC and biomass burning SOC to WSOC obviously increased as pollution aggravated during the haze period in winter. Besides, the contribution of aromatic SOC increased as PM$_{2.5}$ increased in autumn. Therefore, the control of biomass burning and reduction of aromatic precursors would be of great significance for controlling WSOC during the severe haze episodes in winter and autumn.

According to the correlation patterns with the key influencing factors of the gas phase and aqueous phase reactions, the acid-catalyzed heterogeneous or aqueous-phase processing was suggested as the major formation pathway of SOC over the study period in winter, while photooxidation played a critical role during the study period in summer. The photooxidation played a more prominent role in the formation of moderately hydrophilic SOC, whereas heterogeneous or aqueous-phase processing posed more profound effects on the formation of strongly hydrophilic SOC and aging processes of SOC. The SOA modeling based on the chemical transport models has been a powerful tool for SOA study in regional scale. However, SOA modeling remains a challenge and the model studies have shown systematically underestimated SOA concentrations compared to the observed values. The findings of this study would help to reduce the model-measurement discrepancies of SOA by underlining the importance of the SOA properties and the contributions of heterogeneous formation processes in different seasons.

**Data availability.**

The data used in this article are available from the authors upon request (jingchen@bnu.edu.cn).

**Author contribution**

The corresponding author, JC, provided the ideas and funding, discussed the results, and revised the paper, QY conducted the sampling and chemical analysis, analyzed the data, drawn the picture, and wrote the manuscript. WhQ, SmC, YpZ, YwS, KX, MA contributed to the field sampling and put forward suggestions on the discussion. WhQ and YpZ helped with the

data processing from FINNv1.5.

**Competing interests.**

The authors declare that they have no conflict of interest.

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

965

**Table 1** The average concentrations and standard deviations of the identified carbonaceous species in $PM_{2.5}$ during the sampling periods in four seasons.

| Compounds ($\mu g\ m^{-3}$) | Winter | | | Spring | | | Summer | | | Autumn | | |
|---|---|---|---|---|---|---|---|---|---|---|---|---|
| | Daytime | Nighttime | Mean | Daytime | Nighttime | Mean | Daytime | Nighttime | Mean | Daytime | Nighttime | Mean |
| CO | 1.7 ± 1.7 | 2.2 ± 2.8 | 1.9 ± 2.3 | 0.6 ± 0.4[*] | 0.8 ± 0.4[*] | 0.7 ± 0.4 | 1.1 ± 0.4 | 0.9 ± 0.2 | 1.0 ± 0.3 | 1.1 ± 0.4[*] | 1.4 ± 0.4[*] | 1.2 ± 0.4 |
| $PM_{2.5}$ | 120 ± 107 | 147 ± 154 | 133 ± 131 | 60.6 ± 36.2 | 64.5 ± 34.8 | 62.5 ± 34.9 | 59.8 ± 28.6 | 51.9 ± 20.6 | 55.8 ± 24.8 | 75.2 ± 58.1 | 81.1 ± 50.8 | 78.2 ± 53.7 |
| OC | 20.1 ± 19.2 | 21.0 ± 24.8 | 20.6 ± 21.9 | 7.9 ± 2.6 | 9.5 ± 3.4 | 8.7 ± 3.1 | 8.7 ± 3.4[*] | 6.8 ± 4.3[*] | 7.8 ± 3.9 | 9.4 ± 3.8 | 10.1 ± 3.7 | 9.7 ± 3.7 |
| EC | 3.9 ± 3.1 | 4.7 ± 5.8 | 4.3 ± 4.6 | 1.9 ± 1.1[*] | 2.7 ±1.4[*] | 2.3 ± 1.3 | 1.4 ± 1.0 | 1.3 ± 1.0 | 1.3 ± 1.0 | 2.4 ± 1.4[*] | 3.4 ± 1.7[*] | 2.9 ± 1.6 |
| OC/EC | 4.6 ± 1.1 | 4.3 ± 1.3 | 4.5 ± 1.2 | 5.2 ± 2.1 | 4.4 ± 2.5 | 4.8 ± 2.3 | 6.7 ± 3.9 | 5.3 ± 4.2 | 6.1 ± 4.1 | 4.4 ± 1.6[*] | 3.3 ± 0.9[*] | 3.8 ± 1.4 |
| WSOC | 11.4 ± 11.3 | 12.0 ± 16.4 | 11.7 ± 13.9 | 4.1 ± 2.0 | 4.7 ± 2.6 | 4.4 ± 2.3 | 5.3 ± 2.1[*] | 4.0 ± 2.7[*] | 4.7 ± 2.5 | 4.7 ± 3.0 | 4.9 ± 2.8 | 4.8 ± 2.8 |
| WSOC/OC | 0.53 ± 0.08 | 0.51 ± 0.08 | 0.52 ± 0.08 | 0.50 ± 0.10 | 0.47 ± 0.14 | 0.49 ± 0.12 | 0.62 ± 0.11 | 0.59 ± 0.10 | 0.60 ± 0.11 | 0.47 ± 0.12 | 0.46 ± 0.12 | 0.46 ± 0.12 |
| MH-WSOC | 7.9 ± 7.6 | 8.0 ± 10.3 | 8.0 ± 8.9 | 2.8 ± 1.3 | 2.9 ± 1.6 | 2.9 ± 1.5 | 4.1 ± 1.2 | 3.4 ± 1.6 | 3.8 ± 1.5 | 2.9 ± 1.6 | 2.9 ± 1.3 | 2.9 ± 1.5 |
| SH-WSOC | 3.2 ± 3.8 | 4.0 ± 6.1 | 3.6 ± 5.0 | 1.3 ± 0.9 | 1.8 ± 1.1 | 1.6 ± 1.0 | 1.2 ± 1.0[*] | 0.7 ± 1.1[*] | 1.0 ± 1.1 | 1.8 ± 1.4 | 2.0 ± 1.5 | 1.9 ± 1.4 |
| **Organic tracers ($ng\ m^{-3}$)** | | | | | | | | | | | | |
| Levoglucosan | 307 ± 300 | 388 ± 394 | 349 ± 348 | 100 ± 87.8[*] | 194 ± 175[*] | 147 ± 144 | 23.6 ± 11.0 | 34.2 ± 24.2 | 28.9 ± 19.3 | 136 ± 102[*] | 234 ± 125[*] | 185 ± 123 |
| Cholesterol | 5.0 ± 3.0 | 4.9 ± 3.3 | 4.9 ± 3.1 | 3.9 ± 1.9 | 4.8 ± 2.5 | 4.3 ± 2.3 | 4.1 ± 2.4 | 3.0 ± 1.1 | 3.6 ± 1.9 | 6.1 ± 4.4 | 6.3 ± 3.1 | 6.2 ± 3.8 |
| Phthalic acid | 88.7 ± 84.8 | 90.8 ± 121 | 89.8 ± 103 | 27.3 ± 20.8 | 21.9 ± 14.0 | 24.6 ± 17.7 | 55.9 ± 22.0[*] | 17.6 ± 9.1[*] | 36.8 ± 25.5 | 27.6 ± 21.8 | 19.9 ± 13.3 | 23.8 ± 18.2 |
| 4-Methyl-5-nitrocatechol | 24.7 ± 26.4 | 35.2 ± 41.0 | 30.1 ± 34.5 | 1.8 ± 1.9 | 3.3 ± 2.7 | 2.6 ± 2.4 | 0.1 ± 0.3 | 0.0 ± 0.0 | 0.1 ± 0.2 | 1.6 ± 1.2[*] | 4.4 ± 3.6[*] | 3.0 ± 3.0 |
| 2-Methylerythritol | 2.1 ± 2.3 | 2.2 ± 3.5 | 2.2 ± 2.9 | 1.2 ± 0.6 | 1.5 ± 0.8 | 1.4 ± 0.7 | 55.4 ± 48.5 | 41.6 ± 34.6 | 48.5 ± 42.0 | 2.3 ± 1.1 | 2.6 ± 1.2 | 2.5 ± 1.1 |
| 3-Hydroxyglutaric acid | 4.4 ± 3.9 | 4.2 ± 5.0 | 4.3 ± 4.5 | 4.2 ± 2.8 | 4.9 ± 5.1 | 4.6 ± 4.0 | 37.1 ± 22.7[*] | 27.3 ± 18.5[*] | 32.2 ± 20.9 | 7.5 ± 4.6 | 7.0 ± 4.4 | 7.2 ± 4.5 |
| *cis*-Pinonic acid | 3.3 ± 2.4 | 3.0 ± 2.1 | 3.2 ± 2.2 | 9.0 ± 6.0[*] | 6.9 ± 3.6[*] | 7.9 ± 5.0 | 7.3 ± 4.2 | 10.1 ± 6.0 | 8.7 ± 5.3 | 7.3 ± 3.0[*] | 3.6 ± 0.8[*] | 5.5 ± 2.9 |

[*] According to paired t test, the values with [*] showed statistically significant differences ($p<0.05$) between day and night.

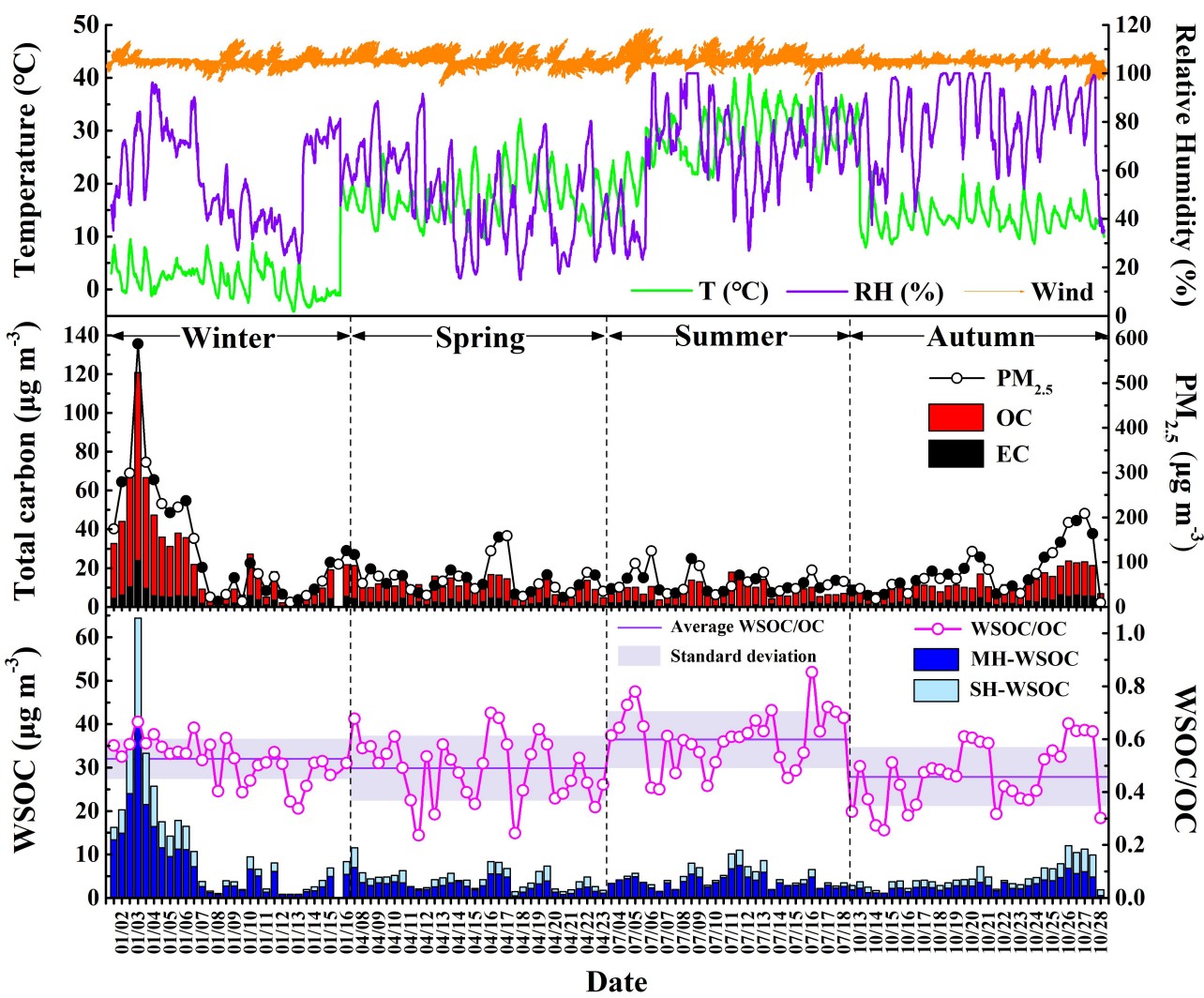

**Figure 1.** Temporal variations of meteorological parameters, the mass concentrations of PM$_{2.5}$, OC, EC, WSOC and WSOC/OC ratio in Beijing during the sampling periods in four seasons of 2017.

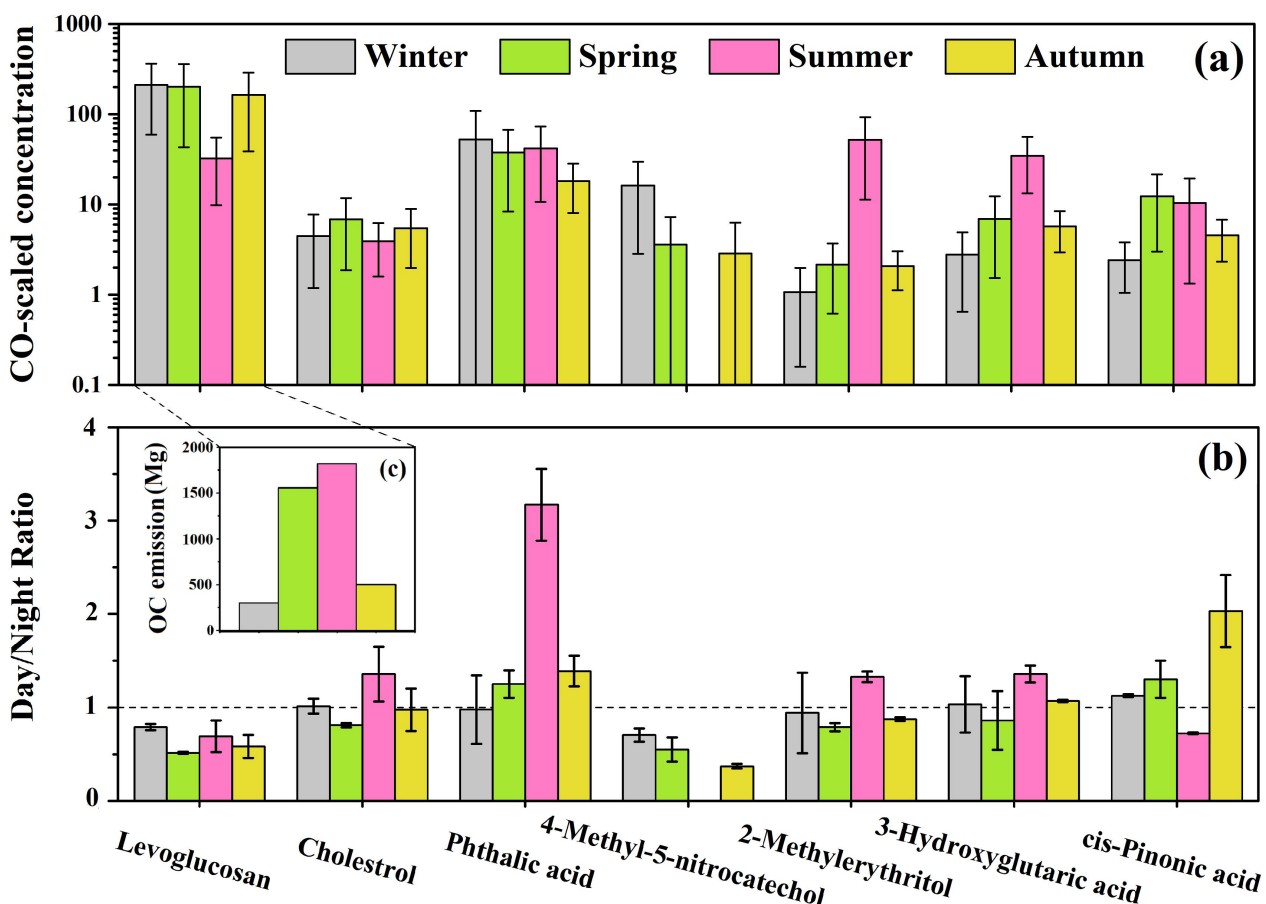

Figure 2. (a) The CO-scaled concentration of the identified organic tracers; (b) the day to night ratios of the measured concentrations of the organic tracers; (c) the OC emission amounts from open biomass burning provided by the Fire Inventory (FINN) in Beijing during the sampling periods in four seasons of 2017.

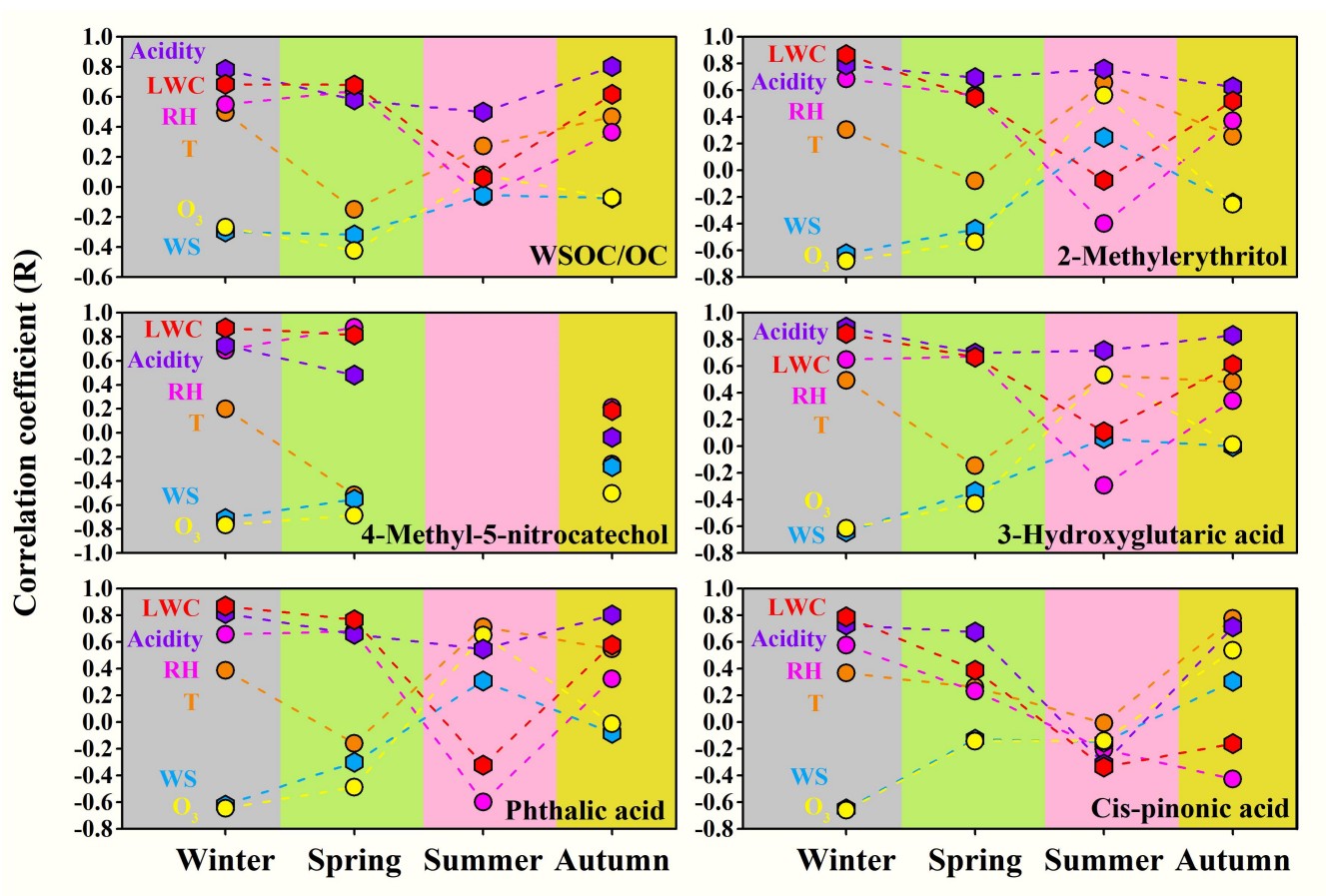

**Figure 3.** Correlation coefficients between SOA tracers and WSOC/OC and the meteorological parameters, $O_3$ concentration, aerosol acidity and LWC during the sampling periods in four seasons of 2017.

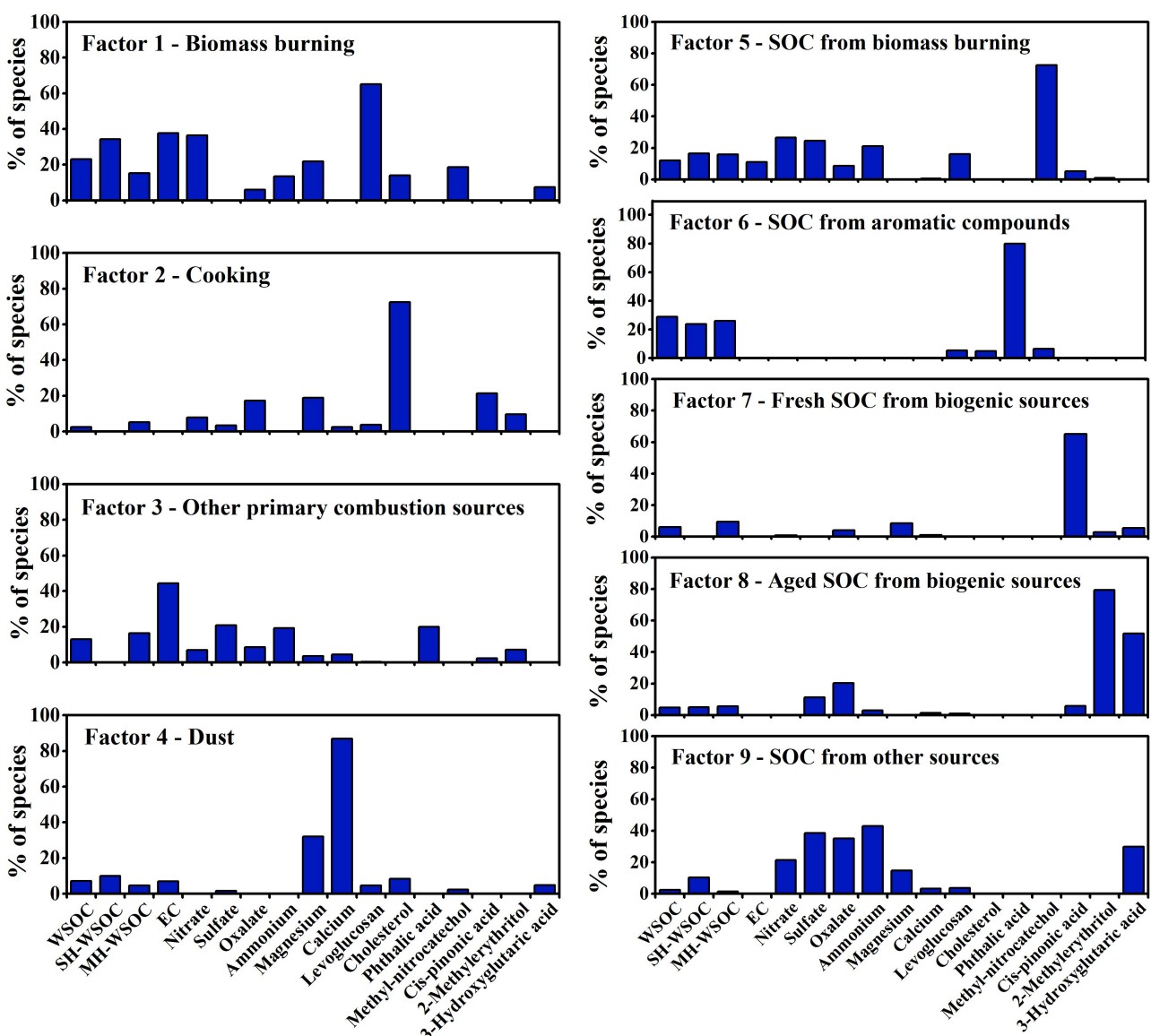

**Figure 4**. The constrained 9-factor solution resolved by the PMF model.

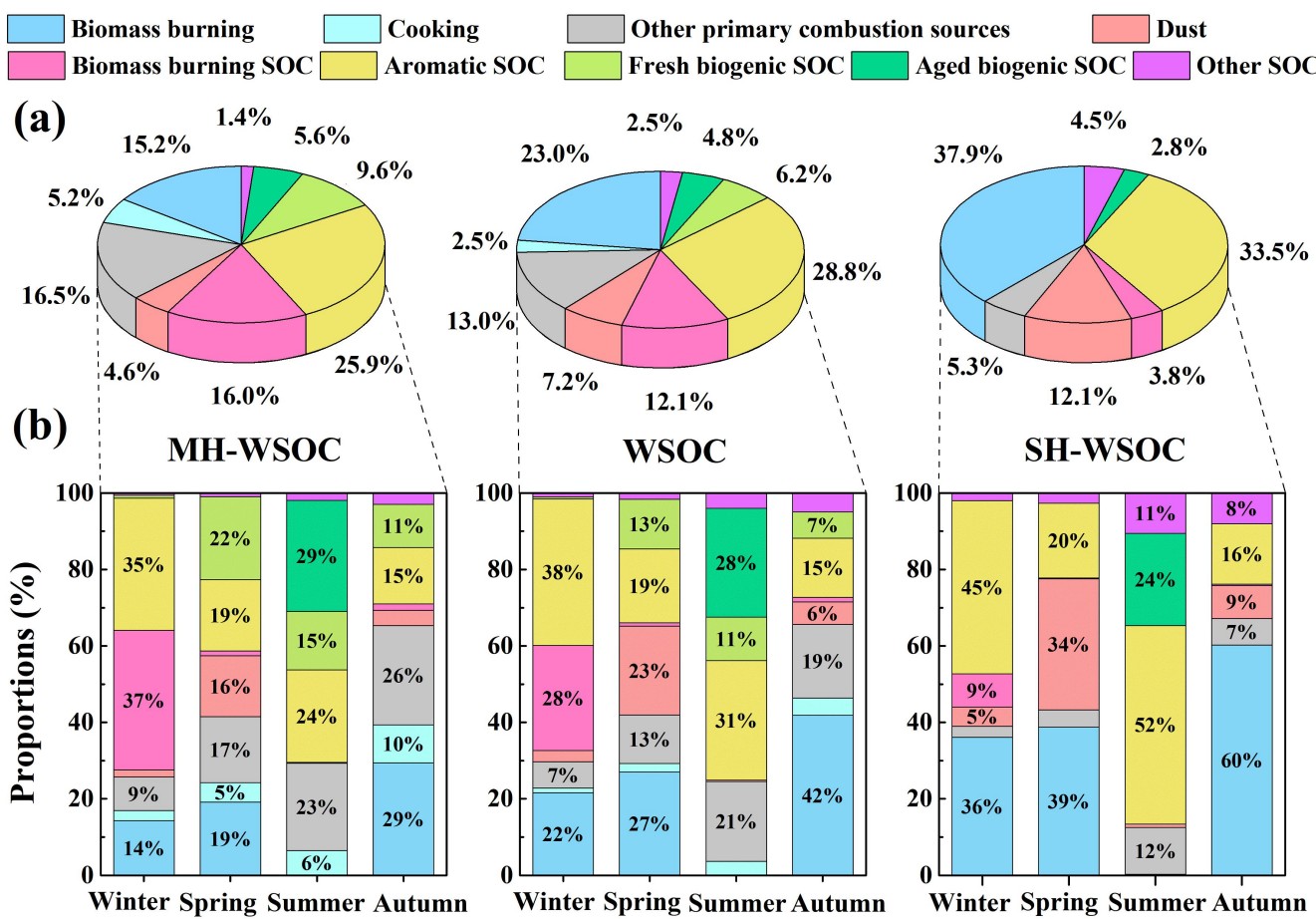

**Figure 5.** Source contributions of WSOC as well as its moderately hydrophilic and strongly hydrophilic fractions in Beijing: (a) comparison of the source contributions to total WSOC, moderately hydrophilic WSOC and strongly hydrophilic WSOC on a whole year scale; (b) seasonal variation of the source contributions to total, moderately and strongly hydrophilic WSOC during the sampling periods in four seasons.

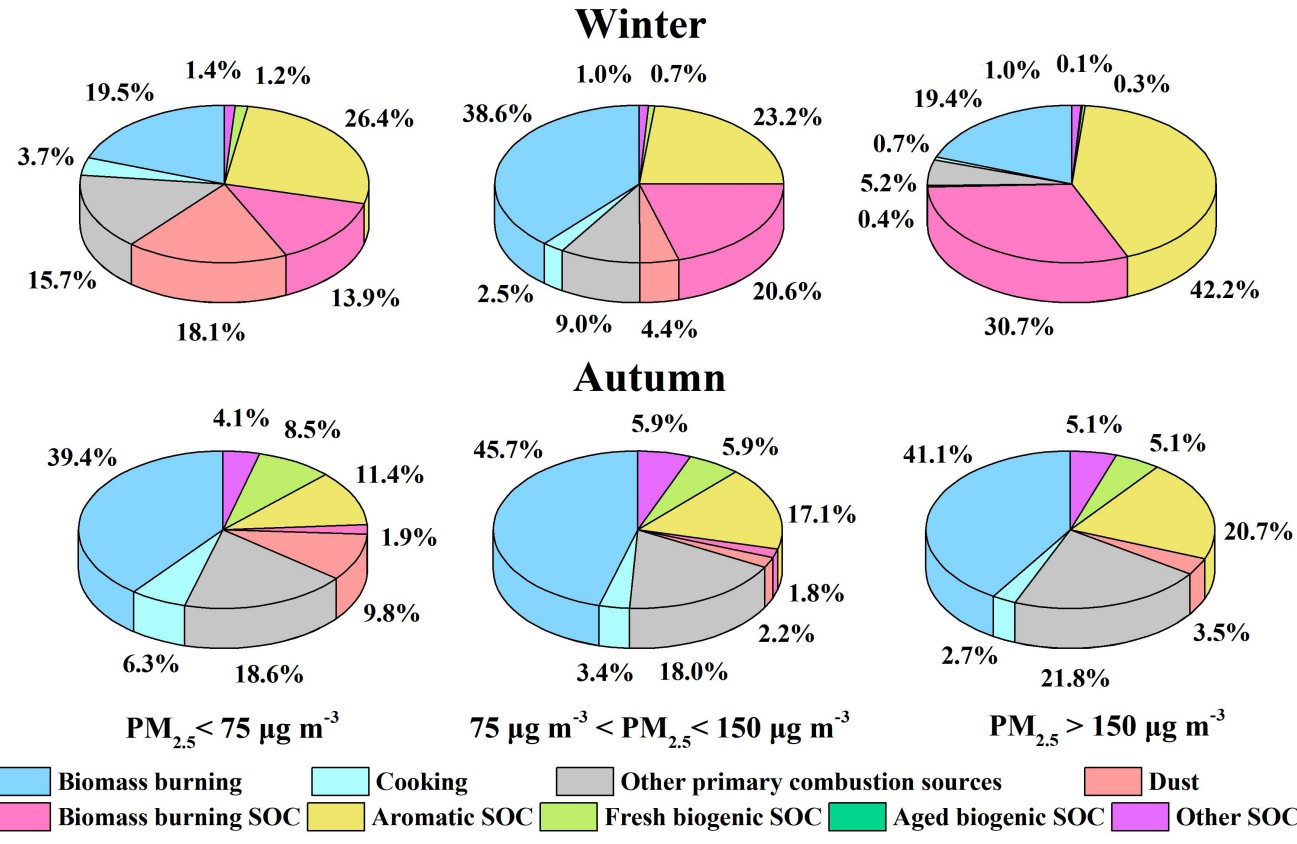

**Figure 6**. Source contributions to WSOC in PM₂.₅ at different pollution levels in winter and autumn.

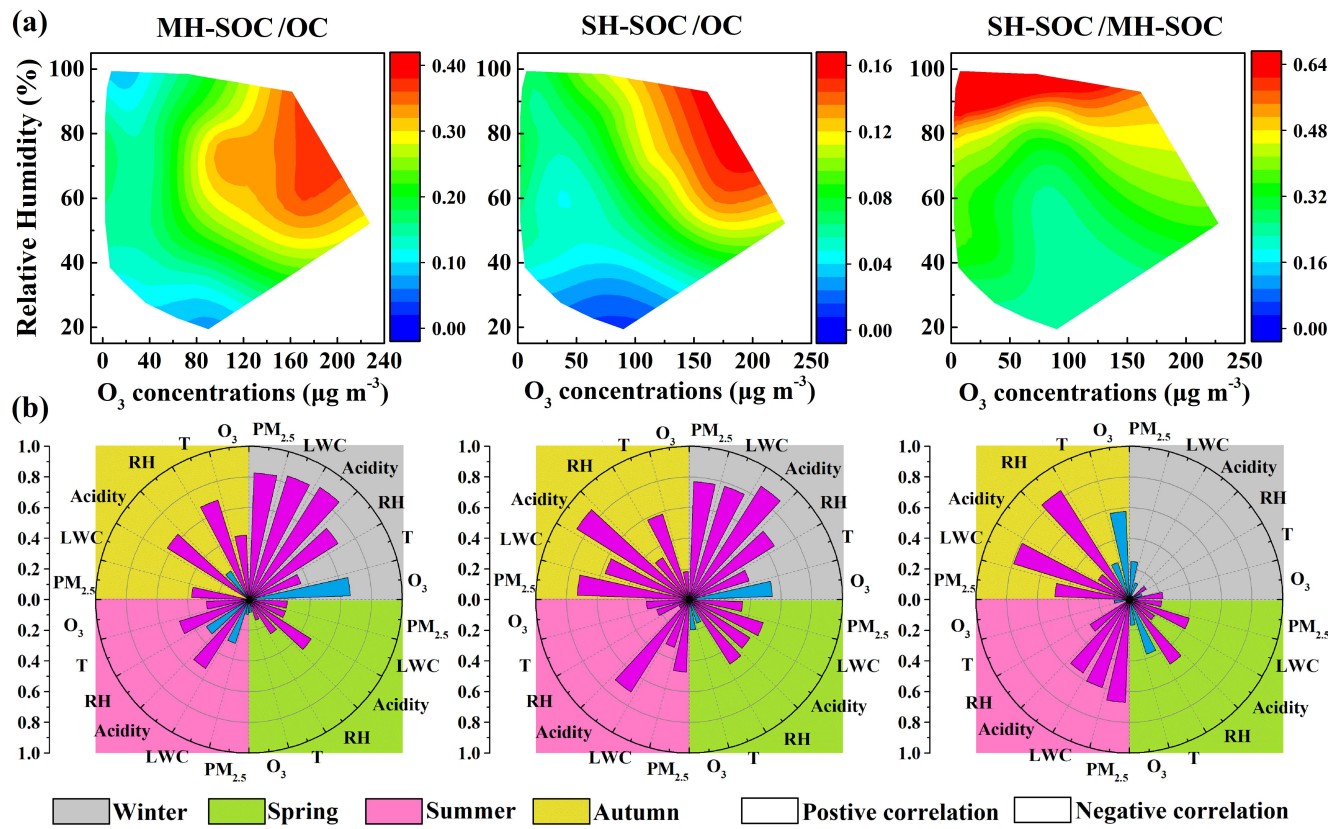

**Figure 7**. (a) The RH versus $O_3$ dependence of MH-SOC/OC, SH-SOC/OC and SH-SOC/MH-SOC ratios during the whole sampling period, and (b) correlation coefficients between the above ratios and meteorological parameters, $O_3$, $PM_{2.5}$, aerosol acidity and LWC in the four seasons.

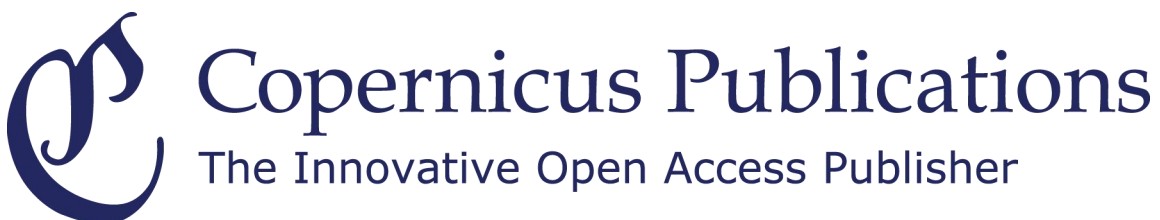

**Figure: The logo of Copernicus Publications.**