# Peer review of "Characteristics, primary sources and secondary formation of water soluble organic aerosols in downtown Beijing"

_Atmospheric Chemistry and Physics, 2020_

## Referee Comment (RC1) · Anonymous Referee #1 · 15 Sep 2020

Yu et al reports observations of organic aerosol, both primary and secondary, collected on filters for different seasons/time periods of 2017 in Beijing, China. They report water soluble organic carbon (WSOC), its hydrophobic and hyrdophilic portions, water soluble ions, total PM2.5, total organic carbon (OC), and total elemental carbon (EC). Further, they report tracers associated to different sources (levoglucosan, cholesterol, phthalic acid, 4-methyl-5-nitrocatechol, 2-methylerythritol, 3-hydroxyglutaric acid, and cis-pinoic acid). They use the tracers to differentiate sources of OC and WSOC during the seasons via "CO-scaled" concentrations, day and night ratios, correlation coefficients with various meteorological and chemical properties of aerosol ("acidity" and liquid water content), and positive matrix factorization. They find that aqueous chem-

istry explains a large portion of the secondary organic carbon during most seasons except summer, where photochemistry explains an important biogenic portion. They also find differences in the sources between the seasons (biomass burning vs dust vs vegetation). Overall, the paper is important and of interest to Atmsopheric Chemistry and Physics community as there is general overall uncertainty in the sources of organic aerosol in urban environments, especially during all seasons and high pollution events. The paper will be of value once the authors address the comments below.

Section 2.1 Sampling: Further information is needed here for the readers to have a better understanding of how the aerosol was collected–Was there a drier in-line prior to be collected on the filters? Was there a denuder to scrub gases prior to the filter to minimize gas-particle partitioning? Was there an impactor or cyclone for size selection?

Further, of importance, was there any analysis of potential reactions that occurred on the filters prior to sampling?

Line 121: It is unclear what the standards curves were of (the tracers reported throughout paper or other standards), and what is meant by "standard curves with five to seven concentration gradients were re-established." What was re-established?

Line 122: What authentic standards? What company/purity?

Section 2.3: Please describe or cite the PMF software used. PMF 5.0 is not enough to understand how positive matrix factorization was actually conducted.

Line 140: I highly recommend the use of "aerosol acidity," as defined in this line, due to the discussion from Pye et al. 2020 (https://acp.copernicus.org/articles/20/4809/2020/). The ratio here does not define acidity, and is analytically challenging to say if it is defining the amount of hydronium ions in the aerosol phase, as the hydronium ions may be a very low detection limits that cannot be quantified due to propagation of uncertainty.

Section 3: Though an important and valuable aspect of this whole manuscript is that the

filters were collected during different seasons, I highly recommend the authors soften the language throughout that the results "reflect" a specific season or are similar or different to other studies. Since it's only for one year and approximately 2 weeks for each season. The limited data makes it hard to say how typical the results are and this should be discussed/emphasized throughout (instead of general statements that in fall this is what is observed/happens).

Another area I suggest the authors be careful in their discussion of r values, as majority of the values they report lead to Rˆ2 values less than 0.5 (thus explaining less than 50% of the variability observed).

Line 195: Since the authors are comparing OC from emissions inventory to Fig. 2, I would recommend converting the emissions to OC-to-CO ratios. Also, I would recommend adding these ratios, if possible, to Fig. 2, for direct comparisons with observations.

Line 199: Is it possible to get emission inventory values of residential biofuel combustion and coal combustion to compare with the OC from open biomass burning?

Line 207: It is unclear how aromatics form SOA to impact WSOC during winter, as the photochemistry is greatly reduced. Could the authors provide more discussion concerning this?

Line 249: It is surprising that the authors are saying that gas-phase photoxidation was not the dominant formation mechanism of secondary organic carbon. I can see maybe WSOC, but seeing all secondary organic carbon is a big statement. Especially, since the authors go on in line 254-55 to say photochemistry plays a role.

Line 339: Source 3 did not show the highest contribution in winter....highest contribution of what?

Line 416-419: I would recommend caution here, as other hypothesis have been stated for reasons in differences between chamber SOA and ambient SOA, including losses

of vapors to the walls and autoxidation (which has been shut down in chamber experiments due to too high NOx levels and/or too high aerosol loadings).

Table 1: I would recommend somehow highlighting which values show statistical differences between day and night and between seasons. Also, I would recommend including average CO mixing ratios.

Fig. 1: I would recommend including a line that shows the average and standard deviation for the WSOC/OC values. Currently, eyeballing the values in Fig. 1, they look fairly similar in all seasons.

Minor: Line 30: replace takes up with composes Line 117: replace entirely dryness with either "entirely dry" or "entire dryness" Line 121: replace T The with The Line 248: replace did not appear any with did not have any Line 280: replace association with correlation Line 294: replace appeared with showed Line 340: replace except with beyond Line 377: remove Nevertheless Line 399: remove of after Both Line 408: believe C is missing after SO Line 427: replace was in consistent with was consistent

---

## Referee Comment (RC2) · Anonymous Referee #2 · 12 Oct 2020

This work reported 4-season filter-based WSOC measurements including tracer measurements and group separation of the aqueous extracts into so-called hydrophobic and hydrophilic fractions by the SPE method. The sources of WSOC were speculated based on some correlations with O3, RH, ALWC etc. The authors also conducted the PMF analysis to evaluate the source contributions. The problem is the quality of data analysis and discussion. Many of the discussions were not logically presented. Loss terms (e.g., photolysis, chemical reactions, gas-particle partitioning) were generally ignored. Conclusions about the relative contributions of photochemical vs aqueous pathways were made mainly on the basis of simple correlations with O3 or ALWC etc., which can be largely uncertain especially for the winter-haze episodes when all com-

ponents of PM2.5 including primary species were correlated with ALWC or RH. There is also a lack of sufficient information to validate the PMF analysis in this study. The presented PMF results seem quite arbitrary.

Specific comments are listed below.

Page 1, Line 14; Page 2, Line 53-57; Page 4, Line 106-112: Different SPE columns and extraction procedures (e.g., pH) result in various fractions of the WSOC (Sullivan et al., 2006). The authors used SPE (Oasis HLB) to separate the "hydrophilic and hydrophobic" fractions of WSOC. However, as described by Kiss et al. (2002), the one-step SPE on Oasis HLB column is to separate the WSOC into moderately hydrophilic (retained on the column) and strongly hydrophilic (passed through the column) fractions. I think it is wrong to simply assign the retained fraction herein as "hydrophobic" or "mainly HULIS" and the passed-through fraction as typical "hydrophilic (short-chain dicarboxylic acids and saccharides)".

Introduction: Previous understanding of the characteristics of WSOC and its separated fractions as well as their primary and secondary sources were poorly summarized in the current Introduction section.

Page 4, Line 97 and Page 5, Line 124-125: How were the field blanks collected before and after sampling? What exactly were corrected?

Page 5, Line 129-131: Detailed information about the PMF analysis should be provided. The authors said that "the uncertainties were calculated referring to the measured RSD data of chemical analysis and previous studies". It is unclear to me whether this is a right approach. What do "the measured RSD data of chemical analysis and previous studies" mean specifically? Also, the authors said "The PMF model was run repeatedly to obtain a clear and reasonable source profile". How? The reasons of the selection of the numbers of PMF factors as well as the PMF uncertainty estimates and diagnostics are necessary.

Page 5, Line 140-141: The equation calculates ion balance not "aerosol acidity".

Page 6, Line 166: Why would the reduction of open biomass burning lead to decreased WSOC/OC ratios? Please clarify and cite references to support the reduction of open biomass burning.

Page 6, Line 173-176: The correlations (r = 0.44-0.58) are not strong. I think it is difficult to conclude that OA became more hygroscopic as pollution aggravated. Indeed, if primary sources make a large contribution, e.g., in winter when coal combustion was enhanced, OA might not be more hygroscopic although its concentration became greater.

Page 6, Line 180: Perhaps remove "ideal". Levoglucosan is not chemically inert. It is also not a unique tracer for biomass burning. As the authors mentioned in Page 7, Line 199, biofuel and coal burning are also sources of levoglucosan.

Page 7, Line 183: Methylnitrocatechol is not necessarily secondary. Wang X et al. (2017) showed primary emissions of methylnitrocatechol from biomass burning. Coal burning is also a source. The statement that "4-methyl-5-nitrocatechol is a good indicator for biomass burning SOA" is perhaps inappropriate.

Page 7, Line 200-203; Line 210-218; Page 8, Line 237-243: Errors can be propagated to the ratios in Figure 2b so that the day/night CO-scaled ratios can be discussed statistically (not just seasonal mean values). The authors said the CO-scaled concentration of cholesterol was close for the whole sampling period. However, panel a is in log scale. I think it is hard to conclude that 4 vs. 7 (i.e., 75% difference) is "close".

Besides the statistical issue, other problems exist for the conclusions made on the basis of day/night ratios. First, biomass burning is not the only source of CO. The <1 day/night CO-scaled ratios of levoglucosan can be simply caused by enhanced CO emissions at night from other sources when the biomass burning contributions were constant. Similarly for cooking. Second, from emissions to concentrations, many factors are involved. Biomass burning is not a local source in Beijing. Similar strength of emissions may lead to different concentrations in Beijing because of the atmospheric aging and dilution during the regional transport process. Also, scaling secondary tracers by CO has complicated meaning, especially for biogenic-related tracers. I can't understand the logic behind Line 236-243. Not to mention that the phthalic acid concentrations may be affected by the OA concentrations due to gas-particle partitioning and the photolysis of 4-methyl-5-nitrocatechol might be significant to affect its daytime concentrations.

Page 7, Line 205-207: The cited reference is only for PAH. How about other aromatic precursors (e.g., single-ring aromatics)? Do those typical SOA precursors show higher concentrations in winter? Please note that oxidation conditions are also important when discussing about the secondary formation potential of SOA. In winter, the oxidant concentrations (e.g., OH) might be lower.

Section 3.2.2 in Page 8-10: Conclusions in this section are generally arbitrary. Although correlations sometimes help diagnostics, connections between O3, ALWC, RH, T and the tracer species as well as WSOC/OC are not simple/obvious. For example, in Line 248-251, it was said that "WSOC/OC did not correlate with O3, suggesting that gas-phase photooxidation was not the dominant formation mechanism of SOC". Why? Do the authors assume that WSOC are SOC and gas-phase photooxidation is equivalent to OǍň3? What about terms other than chemical production in the mass balance (e.g., photolysis, primary contributions, and so on)? Besides, the correlation isn't strong (r = 0.5) when the authors sometimes said "significantly correlated". Such kind of correlations might be used as non-conflict evidence for explaining the formation pathways but definitely insufficient to make any conclusions. A common argument is that in winter Beijing all components of PM2.5 often correlate with RH and ALWC, even for primary OA. It is not surprise that 4-methyl-5-nitrocatechol correlate with RH and ALWC. The study done by Wang L et al. (2018) suggest coal and traffic contributions to 4-methyl-5-nitrocatechol were the dominant sources in northern China.

Page 11, Line 314-327: The interpretation of the PMF factors is over simple here. It looks like the authors intentionally choose a PMF solution that has separate factors for individual tracers. However, how do the tracers correlate each other? (1) Previous studies found that methyl-nitrocatechol correlates with AMS BBOA and levoglucosan (Linuma et al. 2010; Mohr et al. 2013). If the two temporally correlates, the split of the biomass burning factor into two (primary and secondary) may be highly uncertain given the small sample size of this study. (2) Factor 7, 8, and 9 are all associated with biogenic SOA tracers. It was said in Page 10, Line 297-298 that 3-hydroxyglutaric acid correlated strongly with 2-methylerythritol. Then how and why to separate Factor 8 and 9. Is 3-hydroxyglutaric acid a unique tracer for monoterpene SOA? For day and night samples which did not maintain much oxidation process information (meaning first-generation vs multi-generation), I am surprised that there were two monoterpene SOA factors (one is marked by cis-pinonic acid and the other is marked by 3-hydroxyglutaric acid). (3) For Factor 3, the profile has significant contributions of secondary species, is it really primary?

Technical remarks:

Page 7, Line 211: "secondary formation of aromatic SOA" - SOA is secondary.

Page 8, Line 239: "the diurnal patterns were close to 1" – What does this mean?

Page 9, Line 252: LWC has already defined.

References:

1. Kiss, G.; Varga, B.; Galambos, I.; Ganszky, I., Characterization of water-soluble organic matter isolated from atmospheric fine aerosol. J. Geophys. Res. 2002, 107 (D21).

2. Sullivan, A. P.; Weber, R. J., Chemical characterization of the ambient organic aerosol soluble in water: 1. Isolation of hydrophobic and hydrophilic fractions with a XAD-8 resin. J. Geophys. Res. 2006, 111 (D5).

3. Wang, L.; Wang, X.; Gu, R.; Wang, H.; Yao, L.; Wen, L.; Zhu, F.; Wang, W.; Xue, L.; Yang, L.; Lu, K.; Chen, J.; Wang, T.; Zhang, Y.; Wang, W., Observations of fine particulate nitrated phenols in four sites in northern China: concentrations, source apportionment, and secondary formation. Atmos. Chem. Phys. 2018, 18 (6), 4349-4359.

4. Wang, X.; Gu, R.; Wang, L.; Xu, W.; Zhang, Y.; Chen, B.; Li, W.; Xue, L.; Chen, J.; Wang, W., Emissions of fine particulate nitrated phenols from the burning of five common types of biomass. Environ. Pollut. 2017, 230, 405-412.

5. Mohr, C.; Lopez-Hilfiker, F. D.; Zotter, P.; Prévôt, A. S. H.; Xu, L.; Ng, N. L.; Herndon, S. C.; Williams, L. R.; Franklin, J. P.; Zahniser, M. S.; Worsnop, D. R.; Knighton, W. B.; Aiken, A. C.; Gorkowski, K. J.; Dubey, M. K.; Allan, J. D.; Thornton, J. A., Contribution of Nitrated Phenols to Wood Burning Brown Carbon Light Absorption in Detling, United Kingdom during Winter Time. Environ. Sci. Technol. 2013, 47 (12), 6316-6324.

---

## Author Comment (AC1) · 22 Nov 2020

**Response to Reviewer 1**

**Overall comment:** Yu et al reports observations of organic aerosol, both primary and secondary, collected on filters for different seasons/time periods of 2017 in Beijing, China. They report water soluble organic carbon (WSOC), its hydrophobic and hydrophilic portions, water soluble ions, total $PM_{2.5}$, total organic carbon (OC), and total elemental carbon (EC). Further, they report tracers associated to different sources (levoglucosan, cholesterol, phthalic acid, 4-methyl-5-nitrocatechol, 2-methylerythritol, 3-hydroxyglutaric acid, and *cis*-pinonic acid). They use the tracers to differentiate sources of OC and WSOC during the seasons via "CO-scaled" concentrations, day and night ratios, correlation coefficients with various meteorological and chemical properties of aerosol ("acidity" and liquid water content), and positive matrix factorization. They find that aqueous chemistry explains a large portion of the secondary organic carbon during most seasons except summer, where photochemistry explains an important biogenic portion. They also find differences in the sources between the seasons (biomass burning vs dust vs vegetation). Overall, the paper is important and of interest to Atmsopheric Chemistry and Physics community as there is general overall uncertainty in the sources of organic aerosol in urban environments, especially during all seasons and high pollution events. The paper will be of value once the authors address the comments below.

**Response:** We deeply appreciate the reviewer's time and efforts devoted to improving this manuscript. We have provided our responses point by point below each comment, and have carefully revised the paper according to the reviewer's valuable suggestions.

1. Section 2.1 Sampling: Further information is needed here for the readers to have a better understanding of how the aerosol was collected–Was there a drier in-line prior to be collected on the filters? Was there a denuder to scrub gases prior to the filter to minimize gas-particle partitioning? Was there an impactor or cyclone for size selection? Further, of importance, was there any analysis of potential reactions that occurred on the filters prior to sampling?

**Response:** Thanks for the reviewer's comment. Our sampler included a PM$_{2.5}$ impactor, but no in-line drier or denuder was used in this study.

Sampling of organic carbon is accompanied by both positive and negative artifacts. The positive artifact is due to the adsorption of gaseous organics to the sampling filter, and the negative artifact is caused by the evaporation of collected particulate organic carbon. To eliminate the positive artifact, a denuder can be placed upstream the sample filter to remove the gaseous organics by diffusion to the adsorbent surface (Cheng et al., 2009). The use of a denuder in the sampling system has been reported in previous studies (Eatough et al., 1993, 1999; Mader et al., 2001; Matsumoto et al., 2003; Viana et al., 2006; Cheng et al., 2009, 2010, 2012; Kristensen et al., 2016). The use of a denuder may induce a larger negative artifact, however, as the removal of gaseous organics can enhance the evaporation of particulate OC. Thus a backup filter should also be included in the sampling system (Cheng et al., 2009). Besides, the flow rate passing through the denuder was very low in most studies (Matsumoto et al., 2003; Viana et al., 2006; Cheng et al., 2009, 2010, 2012; Kristensen et al., 2016). This might be due to the significantly decreased removal efficiency of the denuder as the air flow rate increased (Cui et al., 1998; Ding et al., 2002). To collect enough samples for the accurate measurement of trace organic species, the flow rate of 1.05 m$^3$ min$^{-1}$ was chosen in this study. The air flow rate of about 1.05 m$^3$ min$^{-1}$ has been frequently used in the field sampling of organic aerosols (Kawamura et al., 2013; Verma et al., 2012, 2015; Li et al., 2018; Ma et al., 2018; Huang et al., 2020). At this flow rate, a denuder with a high removal efficiency is hardly commercially available.

Nevertheless, we were aware of the potential sampling artifacts and attempted to estimate the sampling artifacts of OC based on the literature results. Firstly, the adsorption behavior of OC might vary with meteorological conditions. Besides, the OC fractions with different volatility show different adsorption behavior. Cheng et al. (2015) compared the concentrations of different OC fractions (OC1, OC2, OC3, OC4) on bare quartz filters with those on denuded quartz filters in the four seasons of Beijing, and the results are summarized in Table S1. The contributions of different OC

fractions measured in this study are also listed in Table S1.

**Table S1** The ratio of the OC concentrations on the bare quartz filters to those on the denuded quartz filters in Cheng et al. (2015), as well as the contribution of different OC fractions measured in this study.

| | The ratio of OC on bare quartz filters to denuded quartz filters (Cheng et al., 2015) | | | | The contribution of different OC fractions measured in this study | | | |
| --- | --- | --- | --- | --- | --- | --- | --- | --- |
| | OC1 | OC2 | OC3 | OC4 | OC1 | OC2 | OC3 | OC4 |
| Winter | 1.27 | 1.03 | 1.02 | 1.05 | 10.8 % | 19.6 % | 24.7 % | 44.8 % |
| Spring | 2.05 | 1.05 | 1.00 | 1.01 | 3.9 % | 27.2 % | 43.1 % | 25.7 % |
| Summer | 2.45 | 1.60 | 1.17 | 1.08 | 4.4 % | 37.6 % | 36.0 % | 22.0 % |
| Autumn | 2.08 | 1.05 | 0.99 | 1.01 | 7.9 % | 26.5 % | 40.2 % | 25.3 % |

McDow (1986) systematically investigated the effect of sampling procedure on the OC measurement. The adsorption of organic vapors on bare quartz filters ($C_{postive\ artifact}$) was a function of the sampling duration (t) multiplied by the face velocity (v) as follows:

$$C_{postive\ artifact} = \sum_i \rho_i \frac{1 - e^{-\varepsilon_i vt}}{\varepsilon_i vt} \tag{1}$$

where the face velocity (v, cm s$^{-1}$) is the ratio of the flow rate (cm$^3$ s$^{-1}$) to the sampling area of the filter (cm$^2$), $\rho_i$ is the concentration of adsorptive vapor $i$ (g cm$^{-3}$), and $\varepsilon_i$ is a constant which can be defined as:

$$\varepsilon_i = \frac{1}{l}\left[\beta f(t) t_o \exp\left(\frac{Q_A}{RT}\right) + 1\right] \tag{2}$$

where $l$ is the effective filter thickness. The average thickness of the quartz filter used in this study was 463 μm. The other parameters are all constants.

Hence, it can be calculated that $\varepsilon_i > 1/l > 20$ cm$^{-1}$, and $1 - e^{-\varepsilon vt} \approx 1$. Therefore, the positive artifact ($C_{positive\ artifact}$) is inversely proportional to the product of the sampling duration and the face velocity (v×t). The face velocity of Cheng et al. (2015) was 9.8 cm s$^{-1}$, while that in our study was 47.3 cm s$^{-1}$. The sampling duration of Cheng et al. (2015) was 24 h, while that in our study was 12 h. That is to say, the positive artifact of Cheng et al. (2015) was about 2.4 times higher than that in our study.

Based on the literature results and taking into account all the factors (seasons,

OC fractions, sampling procedure), the contribution of positive artifact to the measured OC was estimated to be 2.3 %, 1.4 %, 9.9 %, and 2.2 % in winter, spring, summer and autumn respectively in this study, which is roughly acceptable.

To further estimate the impact of gas-particle partitioning and potential reactions occurring on the filters, we overlapped two quartz filters and took samples at a flow rate of 1.05 $m^3$ $min^{-1}$ for a duration of 12 h. The organic tracers selected in this study were measured in both filters. The organic tracers on the backup filters typically originate from three sources: (1) adsorption of the organic vapors in the atmosphere; (2) adsorption of the semi-volatile species evaporated from the front filter; (3) secondary formation from the adsorbed organic vapors on the backup filter. Except for *cis*-pinonic acid, the tracer concentrations on the backup filter were all less than 5 % of those on the front filters, while the concentration of *cis*-pinonic acid on the backup filter was 21.6 % of that on the front filter. This result suggested that the sampling procedure in this study might bring some uncertainties for the measurement of *cis*-pinonic acid, and the sampling artifact was not significant for the other organic tracers.

**References**

Cheng, Y., He, K., Duan, F., Zheng, M., Ma, Y., and Tan, J.: Positive sampling artifact of carbonaceous aerosols and its influence on the thermal-optical split of OC/EC, Atmos. Chem. Phys., 9, 7243-7256, 2009.

Cheng, Y., He, K., Duan, F., Zheng, M., Ma, Y., Tan, J., and Du, Z.: Improved measurement of carbonaceous aerosol: evaluation of the sampling artifacts and inter-comparison of the thermal-optical analysis methods, Atmos. Chem. Phys., 10, 8533-8548, doi:10.5194/acp-10-8533-2010, 2010.

Cheng, Y., Duan, F., He, K., Du, Z., Zheng, M., and Ma, Y.: Sampling artifacts of organic and inorganic aerosol: Implications for the speciation measurement of particulate matter, Atmos. Environ., 55, 229-233, doi:10.1016/j.atmosenv.2012.03.032, 2012.

Cheng, Y., and He, K.: Uncertainties in observational data on organic aerosol: An annual perspective of sampling artifacts in Beijing, China, Environ. Pollut., 206, 113-121, doi:10.1016/j.envpol.2015.06.012, 2015.

Cui, W., Eatough, D. J., and Eatough, N. L.: Fine particulate organic material in the Los Angeles Basin-I: Assessment of the high-volume Brigham Young University Organic Sampling System, BIG BOSS, J. Air Waste Manage Assoc., 48, 1024-1037, doi:10.1080/10473289.1998.10463760, 1998.

Ding, Y., Pang, Y., and Eatough, D. J.: High-volume diffusion denuder sampler for the routine monitoring of fine particulate matter: I. design and optimization of the PC-BOSS, Aerosol Sci. Tech., 36, 369–382, 2002.

Eatough, D. J., Wadsworth, A., Eatough, D. A., Crawford, J. W., Hansen, L. D., and Lewis, E. A.: A multiple-system, multichannel diffusion denuder sampler for the determination of fine particulate organic material in the atmosphere, Atmos. Environ., 27A, 1213–1219, 1993.

Eatough, D. J., Obeidi, F., Pang, Y., Ding, Y., Eatough, N. L., and Wilson, W. E.: Integrated and real-time diffusion denuder sampler for $PM_{2.5}$, Atmos. Environ., 33, 2835–2844, 1999.

Kawamura, K., Tachibana, E., Okuzawa, E., Aggarwal, S. G., Kanaya, Y., and Wang, Z.: High abundances of water-soluble dicarboxylic acids, ketocarboxylic acids and α-dicarbonyls in the mountaintop aerosols over the North China Plain during wheat burning season, Atmos. Chem. Phys., 13, 8285-8302, doi:10.5194/acp-13-8285-2013, 2013.

Kristensen, K., Bilde, M., Aalto, P. P., Petaja, T., and Glasius, M.: Denuder/filter sampling of organic acids and organosulfates at urban and boreal forest sites: Gas/particle distribution and possible sampling artifacts, Atmos. Environ., 130, 36-53, doi:10.1016/j.atmosenv.2015.10.046, 2016.

Mader, B. T., Flagan, R. C., and Seinfeld, J. H.: Sampling atmospheric carbonaceous aerosols using a particle trap impactor/denuder sampler, Environ. Sci. Technol., 35, 4857-4867, 2001.

Matsumoto, K., Hayano, T., and Uematsu, M: Positive artifact in the measurement of particulate carbonaceous substances using an ambient carbon particulate monitor, Atmos. Environ., 37, 4713-4717, doi:10.1016/j.atmosenv.2003.07.006, 2003.

McDow, S. R.: The effect of sampling procedures on organic aerosol measurement, Ph.D. dissertation, Oregon Graduate Center, 1987.

Viana, M., Chi, X., Maenhaut, W., Cafmeyer, J., Querol, X., Alastuey, A., Mikuska, P., and Vecera, Z.: Influence of sampling artifacts on measured PM, OC, and EC levels in carbonaceous aerosols in an urban area, Aerosol Sci. Tech., 40, 107-117, 2006.

2. Line 121: It is unclear what the standards curves were of (the tracers reported throughout paper or other standards), and what is meant by "standard curves with five to seven concentration gradients were re-established." What was re-established?

**Response:** Thanks for the reviewer's comment. The standard curves were made by the silylation derivatives of the organic tracers, and the detailed information of these standards is listed in Table S2. "Re-established" means that we measured the derivative products of the standard solutions each time before measuring the ambient samples. We have avoided the unclear expression and revised as follows:

"The authentic standards (Table S2) were dissolved in anhydrous pyridine, and diluted to five to seven different concentrations. Then 100 μL of the standard solutions were reacted with 200 μL silylating reagent (BSTFA: TMCS = 99:1) at 75°C for a duration of 70 min. After cooling down to the ambient temperature, these solutions containing derivative products were diluted to 1 mL with n-hexane, and measured by GC/MS/MS right before the analysis of ambient samples. The $R^2$ of the derivative products were above 0.99, indicating good linearities of these standard curves. "

3. Line 122: What authentic standards? What company/purity?

**Response:** The information of the authentic standards is provided in Table S2 below.

**Table S2** The detailed information of the authentic standards used in this study.

| Authentic standard | Molecular formula | CAS number | Company | Purity |
|---|---|---|---|---|
| Levoglucosan | $C_6H_{10}O_5$ | 498-07-7 | Sigma-Aldrich | 99 % |
| Cholesterol | $C_{27}H_{46}O$ | 57-88-5 | Sigma-Aldrich | 93 % |
| Phthalic acid | $C_8H_6O_4$ | 88-99-3 | Sigma-Aldrich | 99 % |
| 4-Methyl-5-nitrocatechol | $C_7H_7NO_4$ | 68906-21-8 | Toronto Research Chemicals | 98 % |
| 2-Methylerythritol | $C_5H_{12}O_4$ | 58698-37-6 | Sigma-Aldrich | 90 % |
| 3-Hydroxyglutaric acid | $C_5H_8O_5$ | 638-18-6 | Sigma-Aldrich | 95 % |
| cis-Pinonic acid | $C_{10}H_{16}O_3$ | 61826-55-9 | Sigma-Aldrich | 98 % |

4. Section 2.3: Please describe or cite the PMF software used. PMF 5.0 is not enough to understand how positive matrix factorization was actually conducted.

**Response:** Thanks for this kind suggestion. We have added a brief introduction to the PMF software in the manuscript. Besides, we have added the detailed information on the uncertainty calculation of the input data, the selection criteria for the optimal solution, the diagnostic plots and error estimation in the supplementary material.

5. Line 140: I highly recommend the use of "aerosol acidity," as defined in this line, due to the discussion from Pye et al. 2020 (https://acp.copernicus.org/articles/20/4809/2020/). The ratio here does not define acidity, and is analytically challenging to say if it is defining the amount of hydronium

ions in the aerosol phase, as the hydronium ions may be a very low detection limits that cannot be quantified due to propagation of uncertainty.

**Response:** We deeply appreciate the reviewer's valuable suggestion. According to Pye et al. (2020), we changed the ratio of $R_{A/C}$ to the approximate value of aerosol pH ($pH_F$) to denote the aerosol acidity. The $pH_F$ value was estimated using the ISORROPIA-II model. And the molality of $H^+$ ($m_{H^+}$), which was calculated by $m_{H^+} = 10^{-pH_F}$, was used instead for the correlation analysis in the revised manuscript.

6. Section 3: Though an important and valuable aspect of this whole manuscript is that the filters were collected during different seasons, I highly recommend the authors soften the language throughout that the results "reflect" a specific season or are similar or different to other studies. Since it's only for one year and approximately 2 weeks for each season. The limited data makes it hard to say how typical the results are and this should be discussed/emphasized throughout (instead of general statements that in fall this is what is observed/happens).

**Response:** Thanks for the reviewer's valuable suggestion. We have changed the terms of "winter/spring/summer/autumn" to "January/April/July/October" or "during the sampling periods in winter/spring/summer/autumn" throughout the revised manuscript, in order to be more specific.

7. Another area I suggest the authors be careful in their discussion of r values, as majority of the values they report lead to R^2 values less than 0.5 (thus explaining less than 50% of the variability observed).

**Response:** We agree with the reviewer that most of the r values led to $R^2$ less than 0.5, thus could only explain less than 50 % of the variability observed, and were insufficient to reach clear conclusions. Therefore, in the revised manuscript, we only explained the r results and avoided making definite conclusions based on the r values.

8. Line 195: Since the authors are comparing OC from emissions inventory to Fig. 2, I would recommend converting the emissions to OC-to-CO ratios. Also, I would

recommend adding these ratios, if possible, to Fig. 2, for direct comparisons with observations.

**Response:** We are deeply grateful for the reviewer's valuable suggestion. As suggested by the reviewer, we have added the emissions of OC from the Fire Inventory to Figure 2 in the revised manuscript. As a tracer for biomass burning, levoglucosan in the atmosphere may derive from both residential biofuel burning and open biomass burning. To determine whether open biomass burning was the dominant type of biomass burning in Beijing, the seasonal variation of the CO-scaled concentration of levoglucosan was compared with that of the OC emission amount from the Fire Inventory in this study. The CO-scaled concentration of levoglucosan showed a totally different seasonal variation trend from that of the OC emission from open biomass burning, therefore, we speculated that open biomass burning was not the dominant category of biomass burning in Beijing. After careful consideration, we thought that it was not necessary to convert the emissions to OC-to-CO ratios. Instead, the absolute emission amounts of OC from the Fire Inventory were added.

[Figure]

**Figure 2.** (a) The CO-scaled concentration of the identified organic tracers; (b) The day to night ratios of the measured concentrations of the organic tracers; (c) The OC emission amounts from open biomass burning provided by the Fire Inventory (FINN)

in Beijing during the sampling periods in four seasons of 2017.

9. Line 199: Is it possible to get emission inventory values of residential biofuel combustion and coal combustion to compare with the OC from open biomass burning?

**Response:** The total emission amounts of OC from the residential sources in January, April, July and October of 2016 can be obtained from the multi-resolution emission inventory for China (MEIC, http://www.MEICmodel.org). However, the residential OC from MEIC is a total amount of biofuel combustion, coal combustion, other fossil fuel combustion, etc. Unfortunately, the detailed OC emission amount from each residential source is not available. Therefore, the respective OC emission inventory values of residential biofuel combustion and coal combustion were not provided in the manuscript.

10. Line 207: It is unclear how aromatics form SOA to impact WSOC during winter, as the photochemistry is greatly reduced. Could the authors provide more discussion concerning this?

**Response:** We deeply appreciate the reviewer's valuable suggestion. The following discussion has been added in the revised manuscript.

Although the concentrations of oxidants were usually lower in winter due to the weaker solar radiation, a previous observation found that the ·OH concentration in Beijing was significantly higher than that in New York, Birmingham and Tokyo, and was nearly 1 order of magnitude higher than that predicted by global models in northern China in winter (Tan et al., 2018). Zhang et al. (2019) indicated that HONO, which was mainly from the heterogeneous reactions of $NO_2$ and traffic emissions, was the major precursor of ·OH in winter. According to the WRF-Chem model simulation, HONO resulted in a significant enhancement (5-25 $\mu g\ m^{-3}$) of SOA formation (most of which were from the aromatic precursors) during a haze episode in winter in the Beijing-Tianjin-Hebei region (Zhang et al., 2019). Besides, some recent studies suggested that the brown carbon-derived singlet molecular oxygen ($^1O_2^*$) in aerosol

liquid water could react rapidly with the electron-rich organics such as PAHs, thus facilitate the aromatic SOA formation (Kaur et al., 2019; Manfrin et al., 2019). This process might be more significant in winter, when the concentration of HULIS was much higher than that in other seasons. Therefore, both the enhanced levels of oxidants including $\cdot OH$ and $^1O_2^*$ and the higher concentrations of aromatic precursors in winter contributed to the enhanced aromatic SOA formation during the study period in winter.

**References**

Tan, Z., Rohrer, F., Lu, K., Ma, X., Bohn, B., Broch, S., Dong, H., Fuchs, H., Gkatzelis, G. I., Hofzumahaus, A., Holland, F., Li, X., Liu, Y., Liu, Y., Novelli, A., Shao, M., Wang, H., Wu, Y., Zeng, L., Hu, M., Kiendler-Scharr, A., Wahner, A., and Zhang, Y.: Wintertime photochemistry in Beijing: observations of $RO_x$ radical concentrations in the North China Plain during the BEST-ONE campaign, Atmos. Chem. Phys., 18, 12391-12411, doi:10.5194/acp-18-12391-2018, 2018.

Zhang, J., Chen, J., Xue, C., Chen, H., Zhang, Q., Liu, X., Mu, Y., Guo, Y., Wang, D., Chen, Y., Li, J., Qu, Y., and An, J.: Impacts of six potential HONO sources on $HO_x$ budgets and SOA formation during a wintertime heavy haze period in the North China Plain, Sci. Total Environ., 681, 110-123, doi:10.1016/j.scitotenv.2019.05.100, 2019.

11. Line 249: It is surprising that the authors are saying that gas-phase photooxidation was not the dominant formation mechanism of secondary organic carbon. I can see maybe WSOC, but seeing all secondary organic carbon is a big statement. Especially, since the authors go on in line 254-55 to say photochemistry plays a role.

**Response:** We deeply appreciate the reviewer's valuable comment. We agree with the reviewer that it was not proper to conclude that "gas-phase photooxidation was not the dominant formation mechanism of SOC" merely based on the result that "WSOC/OC did not show any significant positive correlation with $O_3$ concentrations". The corresponding statement has been deleted in the revised manuscript. Besides, "SOC" has been corrected to "SOC in WSOC" to be more specific.

12. Line 339: Source 3 did not show the highest contribution in winter....highest contribution of what?

**Response:** We feel sorry for the unclear expression. This sentence has been revised as: "The primary emission strength of coal combustion was the strongest in winter among four seasons, since the domestic heating activities required extra amounts of coal combustion in this season. However, the contribution of Factor 3 to WSOC during the study period in winter was not the highest among four seasons, implying that there could be other sources beyond coal combustion included in Factor 3.

13. Line 416-419: I would recommend caution here, as other hypothesis have been stated for reasons in differences between chamber SOA and ambient SOA, including losses of vapors to the walls and autoxidation (which has been shut down in chamber experiments due to too high $NO_x$ levels and/or too high aerosol loadings).

**Response:** We deeply appreciate the reviewer's valuable suggestion. This statement has been revised as: "Previous observation suggested that a large fraction of ambient SOA was more oxidized than those generated in the dry smog chambers, where SOA could only be produced through gas-phase oxidation (Aiken et al., 2008). There have been some hypothesis for the difference between the chamber SOA and ambient SOA, such as the losses of vapors to the walls, and the autoxidation due to the uncertainties in chamber radical environment (McVay et al., 2016; Thornto et al., 2020). Besides, the results of this study also indicated that the aqueous-phase processing, which can produce more hydrophilic SOA, may be one of the reasons for the discrepancy in the oxidation degrees of ambient SOA and chamber SOA (Ervens et al., 2011)."

**References**

McVay, R.C., Zhang, X., Aumont, B., Valorso, R., Camredon, M., La, Y. S., Wennberg, P. O., and Seinfeld, J. H.: SOA formation from the photooxidation of α-pinene: systematic exploration of the simulation of chamber data, Atmos. Chem. Phys., 16, 2785-2802, doi:10.5194/acp-16-2785-2016, 2016.

Thornton, J. A., Shilling, J. E., Shrivastava, M., D'Ambro, E. L., Zawadowicz, M. A., and Liu, J.: A near-explicit mechanistic evaluation of isoprene photochemical secondary organic aerosol formation and evolution: simulations of multiple chamber experiments with and without added $NO_x$, ACS Earth Space Chem., 4, 1161-1181, doi: 10.1021/acsearthspacechem.0c00118, 2020.

14. Table 1: I would recommend somehow highlighting which values show statistical

differences between day and night and between seasons. Also, I would recommend including average CO mixing ratios.

**Response:** We deeply appreciate the reviewer's suggestion. The average values which showed statistical differences among seasons and between day and night have been highlighted as follows, and the average CO concentrations have also been included in Table 1 in the revised manuscript.

**Table 1** The average concentrations and standard deviations of the identified carbonaceous species in $PM_{2.5}$ during the sampling periods in four seasons.

| Compounds (µg m$^{-3}$) | Winter | | | Spring | | | Summer | | | Autumn | | |
|---|---|---|---|---|---|---|---|---|---|---|---|---|
| | Daytime | Nighttime | Mean | Daytime | Nighttime | Mean | Daytime | Nighttime | Mean | Daytime | Nighttime | Mean |
| CO | 1.7 ± 1.7 | 2.2 ± 2.8 | 1.9 ± 2.3$^a$ | 0.6 ± 0.4$^*$ | 0.8 ± 0.4$^*$ | 0.7 ± 0.4$^b$ | 1.1 ± 0.4 | 0.9 ± 0.2 | 1.0 ± 0.3$^b$ | 1.1 ± 0.4$^*$ | 1.4 ± 0.4$^*$ | 1.2 ± 0.4$^b$ |
| PM$_{2.5}$ | 120 ± 107 | 147 ± 154 | 133 ± 131$^a$ | 60.6 ± 36.2 | 64.5 ± 34.8 | 62.5 ± 34.9$^b$ | 59.8 ± 28.6 | 51.9 ± 20.6 | 55.8 ± 24.8$^b$ | 75.2 ± 58.1 | 81.1 ± 50.8 | 78.2 ± 53.7$^b$ |
| OC | 20.1 ± 19.2 | 21.0 ± 24.8 | 20.6 ± 21.9$^a$ | 7.9 ± 2.6 | 9.5 ± 3.4 | 8.7 ± 3.1$^b$ | 8.7 ± 3.4$^*$ | 6.8 ± 4.3$^*$ | 7.8 ± 3.9$^b$ | 9.4 ± 3.8 | 10.1 ± 3.7 | 9.7 ± 3.7$^b$ |
| EC | 3.9 ± 3.1 | 4.7 ± 5.8 | 4.3 ± 4.6$^a$ | 1.9 ± 1.1$^*$ | 2.7 ± 1.4$^*$ | 2.3 ± 1.3$^{b, c}$ | 1.4 ± 1.0 | 1.3 ± 1.0 | 1.3 ± 1.0$^c$ | 2.4 ± 1.4$^*$ | 3.4 ± 1.7$^*$ | 2.9 ± 1.6$^b$ |
| OC/EC | 4.6 ± 1.1 | 4.3 ± 1.3 | 4.5 ± 1.2$^{a, b}$ | 5.2 ± 2.1 | 4.4 ± 2.5 | 4.8 ± 2.3$^{a, b}$ | 6.7 ± 3.9 | 5.3 ± 4.2 | 6.1 ± 4.1$^a$ | 4.4 ± 1.6$^*$ | 3.3 ± 0.9$^*$ | 3.8 ± 1.4$^b$ |
| WSOC | 11.4 ± 11.3 | 12.0 ± 16.4 | 11.7 ± 13.9$^a$ | 4.1 ± 2.0 | 4.7 ± 2.6 | 4.4 ± 2.3$^b$ | 5.3 ± 2.1$^*$ | 4.0 ± 2.7$^*$ | 4.7 ± 2.5$^b$ | 4.7 ± 3.0 | 4.9 ± 2.8 | 4.8 ± 2.8$^b$ |
| WSOC/OC | 0.53 ± 0.08 | 0.51 ± 0.08 | 0.52 ± 0.08$^b$ | 0.50 ± 0.10 | 0.47 ± 0.14 | 0.49 ± 0.12$^b$ | 0.62 ± 0.11 | 0.59 ± 0.10 | 0.60 ± 0.11$^a$ | 0.47 ± 0.12 | 0.46 ± 0.12 | 0.46 ± 0.12$^b$ |
| MH-WSOC | 7.9 ± 7.6 | 8.0 ± 10.3 | 8.0 ± 8.9$^a$ | 2.8 ± 1.3 | 2.9 ± 1.6 | 2.9 ± 1.5$^b$ | 4.1 ± 1.2 | 3.4 ± 1.6 | 3.8 ± 1.5$^b$ | 2.9 ± 1.6 | 2.9 ± 1.3 | 2.9 ± 1.5$^b$ |
| SH-WSOC | 3.2 ± 3.8 | 4.0 ± 6.1 | 3.6 ± 5.0$^a$ | 1.3 ± 0.9 | 1.8 ± 1.1 | 1.6 ± 1.0$^b$ | 1.2 ± 1.0$^*$ | 0.7 ± 1.1$^*$ | 1.0 ± 1.1$^b$ | 1.8 ± 1.4 | 2.0 ± 1.5 | 1.9 ± 1.4$^b$ |
| **Organic tracers (ng m$^{-3}$)** | | | | | | | | | | | | |
| Levoglucosan | 307 ± 300 | 388 ± 394 | 349 ± 348$^a$ | 100 ± 87.8$^*$ | 194 ± 175$^*$ | 147 ± 144$^b$ | 23.6 ± 11.0 | 34.2 ± 24.2 | 28.9 ± 19.3$^c$ | 136 ± 102$^*$ | 234 ± 125$^*$ | 185 ± 123$^b$ |
| Cholesterol | 5.0 ± 3.0 | 4.9 ± 3.3 | 4.9 ± 3.1$^{a, b}$ | 3.9 ± 1.9 | 4.8 ± 2.5 | 4.3 ± 2.3$^b$ | 4.1 ± 2.4 | 3.0 ± 1.1 | 3.6 ± 1.9$^b$ | 6.1 ± 4.4 | 6.3 ± 3.1 | 6.2 ± 3.8$^a$ |
| Phthalic acid | 88.7 ± 84.8 | 90.8 ± 121 | 89.8 ± 103$^a$ | 27.3 ± 20.8 | 21.9 ± 14.0 | 24.6 ± 17.7$^b$ | 55.9 ± 22.0$^*$ | 17.6 ± 9.1$^*$ | 36.8 ± 25.5$^b$ | 27.6 ± 21.8 | 19.9 ± 13.3 | 23.8 ± 18.2$^b$ |
| 4-Methyl-5-nitrocatechol | 24.7 ± 26.4 | 35.2 ± 41.0 | 30.1 ± 34.5$^a$ | 1.8 ± 1.9 | 3.3 ± 2.7 | 2.6 ± 2.4$^b$ | 0.1 ± 0.3 | 0.0 ± 0.0 | 0.1 ± 0.2$^b$ | 1.6 ± 1.2$^*$ | 4.4 ± 3.6$^*$ | 3.0 ± 3.0$^b$ |
| 2-Methylerythritol | 2.1 ± 2.3 | 2.2 ± 3.5 | 2.2 ± 2.9$^b$ | 1.2 ± 0.6 | 1.5 ± 0.8 | 1.4 ± 0.7$^b$ | 55.4 ± 48.5 | 41.6 ± 34.6 | 48.5 ± 42.0$^a$ | 2.3 ± 1.1 | 2.6 ± 1.2 | 2.5 ± 1.1$^b$ |
| 3-Hydroxyglutaric acid | 4.4 ± 3.9 | 4.2 ± 5.0 | 4.3 ± 4.5$^b$ | 4.2 ± 2.8 | 4.9 ± 5.1 | 4.6 ± 4.0$^b$ | 37.1 ± 22.7$^*$ | 27.3 ± 18.5$^*$ | 32.2 ± 20.9$^a$ | 7.5 ± 4.6 | 7.0 ± 4.4 | 7.2 ± 4.5$^b$ |
| *cis*-Pinonic acid | 3.3 ± 2.4 | 3.0 ± 2.1 | 3.2 ± 2.2$^c$ | 9.0 ± 6.0$^*$ | 6.9 ± 3.6$^*$ | 7.9 ± 5.0$^a$ | 7.3 ± 4.2 | 10.1 ± 6.0 | 8.7 ± 5.3$^a$ | 7.3 ± 3.0$^*$ | 3.6 ± 0.8$^*$ | 5.5 ± 2.9$^b$ |

$^{a, b, c}$ We performed one-way analysis of variance (ANOVA) to evaluate whether these mean values showed statistically significant differences (p<0.05) between two seasons. If two seasonal average values have one or more same superscripts, it means that they did not show significant differences (p>0.05). In contrast, if two average values do not have any same superscript, it means that they showed significant differences (p<0.05). For example, the PM$_{2.5}$ concentration was significantly higher in winter than in other seasons, but it did not show significant difference between spring and summer, spring and autumn, or summer and autumn. Besides, the EC concentration did not show significant difference between spring and summer, as well as between spring and autumn, but it showed significant difference between summer and autumn.

$^*$ We also performed paired t test to evaluate whether daytime and nighttime values showed statistically significant differences (p<0.05). The values with $^*$ as their superscripts showed statistically significant differences between day and night.

15. Fig. 1: I would recommend including a line that shows the average and standard deviation for the WSOC/OC values. Currently, eyeballing the values in Fig. 1, they look fairly similar in all seasons.

**Response:** We deeply appreciate this nice suggestion. As suggested by the reviewer, we have added four lines which represented the average WSOC/OC values in each season, and four shaded areas that showed the standard deviations of WSOC/OC values.

[Figure]

**Figure 1**. Temporal variations of meteorological parameters, the mass concentrations of PM$_{2.5}$, OC, EC, WSOC and WSOC/OC ratio in Beijing during the sampling periods in four seasons of 2017.

16. Minor: Line 30: replace "takes up" with "composes";

Line 117: replace "entirely dryness" with either "entirely dry" or "entire dryness";

Line 121: replace "T The" with "The";

Line 248: replace "did not appear any" with "did not have any";

Line 280: replace "association" with "correlation";

Line 294: replace "appeared" with "showed";

Line 340: replace "except" with "beyond";

Line 377: remove "Nevertheless";

Line 399: remove "of" after Both;

Line 408: believe "C" is missing after "SO";

Line 427: replace "was in consistent" with "was consistent".

**Response:** We deeply appreciate the reviewer's careful and detailed comments. We have corrected these errors in the revised manuscript. Thank you very much!

---

## Author Comment (AC2) · 22 Nov 2020

**Response to Reviewer 2**

**Overall comment:** This work reported 4-season filter-based WSOC measurements including tracer measurements and group separation of the aqueous extracts into so-called hydrophobic and hydrophilic fractions by the SPE method. The sources of WSOC were speculated based on some correlations with O3, RH, ALWC etc. The authors also conducted the PMF analysis to evaluate the source contributions. The problem is the quality of data analysis and discussion. Many of the discussions were not logically presented. Loss terms (e.g., photolysis, chemical reactions, gas-particle partitioning) were generally ignored. Conclusions about the relative contributions of photochemical vs aqueous pathways were made mainly on the basis of simple correlations with O3 or ALWC etc., which can be largely uncertain especially for the winter-haze episodes when all components of PM2.5 including primary species were correlated with ALWC or RH. There is also a lack of sufficient information to validate the PMF analysis in this study. The presented PMF results seem quite arbitrary.

**Response:** We deeply appreciate the reviewer's rigorous consideration and valuable comments, which enabled us to essentially improve the analysis and interpretation of the data. We have fully considered and carefully addressed all the comments raised by the reviewer, and have thoroughly revised the manuscript accordingly. The potential effect of photolysis, chemical reactions and gas-particle partitioning have been discussed in the revised manuscript. The relative contributions of photochemical vs aqueous pathways have been more cautiously discussed in the revised manuscript, and the correlation analysis between the ratio of SOA tracers to OC with RH or ALWC has been added to further support some of our speculations. Moreover, for the PMF analysis, we have provided detailed information on the uncertainty calculation of the input data, the selection criteria for the optimal PMF solution, the diagnostic plots and the error estimation of the PMF results in the revised manuscript. Below the comments are our responses point by point, and the revisions have been indicated in the revised manuscript.

**Comment 1**: Page 1, Line 14; Page 2, Line 53-57; Page 4, Line 106-112: Different SPE columns and extraction procedures (e.g., pH) result in various fractions of the WSOC (Sullivan et al., 2006). The authors used SPE (Oasis HLB) to separate the "hydrophilic and hydrophobic" fractions of WSOC. However, as described by Kiss et al. (2002), the one-step SPE on Oasis HLB column is to separate the WSOC into moderately hydrophilic (retained on the column) and strongly hydrophilic (passed through the column) fractions. I think it is wrong to simply assign the retained fraction herein as "hydrophobic" or "mainly HULIS" and the passed-through fraction as typical "hydrophilic (short-chain dicarboxylic acids and saccharides)".

**Response:** We deeply appreciate the reviewer's valuable comment. We used the same SPE column and separating procedure as described by Kiss et al. (2002) to separate different portions of WSOC in this study. As suggested by the reviewer, we have changed the term "hydrophobic WSOC" to "moderately hydrophilic WSOC", and "hydrophilic WSOC" to "strongly hydrophilic WSOC" throughout the revised manuscript.

To correctly identify different portions of WSOC, we have also reviewed the literature for support. As concluded by Kiss et al. (2002), the moderately hydrophilic WSOC (retained on the Oasis HLB column) is composed of humic-like substances (HULIS). This method is commonly used for the determination of atmospheric HULIS (Lin et al., 2010; Lin and Yu, 2011; Fan et al., 2016; Ma et al., 2018; Sengupta et al., 2018). Lin et al. (2010) suggested that several anhydrosugars (levoglucosan, xylose, sucrose) and short-chain organic acids (oxalic acid, succinic acid, malic acid) were present in the passed-through fraction. Therefore, we thought that it should be proper to assign the retained WSOC fraction as "mainly HULIS", and the passed-through portion as "strongly hydrophilic WSOC (short-chain dicarboxylic acids and saccharides)".

In Line 54-57, the references we cited (Verma et al., 2012, 2015; Yu et al., 2018) all used C-18 silica gel SPE columns to separate different WSOC fractions. Varga et al. (2001) compared the performance of C-18 silica gel columns and Oasis HLB columns, and suggested that Oasis HLB columns retained the same fraction of organic

matter as silica-based columns. Hence, the statement that "previous studies have revealed that the moderately and strongly hydrophilic fractions of WSOC show significantly different intrinsic oxidative potential, thus would pose different effects on human health" should be proper here.

**Response:** Thanks for the reviewer's careful concern. We have clarified the corresponding statement as follows.

The field blanks were collected before and after each sampling period, and a total of 8 field blank samples were obtained. The blank filters were put on the filter holder of the PM2.5 sampler without pumping for 1 min, then stored and analyzed together with the ambient samples. Since two field blanks were obtained during each sampling period, the average concentrations of the targeted compounds on these two field

blanks were used to correct the measured concentrations of the ambient samples in the corresponding season. Therefore, the reported concentrations of the targeted species were the measured concentrations on each ambient sample minus the average concentration on the two field blank samples.

**Comment 4:** Page 5, Line 129-131: Detailed information about the PMF analysis should be provided. The authors said that "the uncertainties were calculated referring to the measured RSD data of chemical analysis and previous studies". It is unclear to me whether this is a right approach. What do "the measured RSD data of chemical analysis and previous studies" mean specifically? Also, the authors said "The PMF model was run repeatedly to obtain a clear and reasonable source profile". How? The reasons of the selection of the numbers of PMF factors as well as the PMF uncertainty estimates and diagnostics are necessary.

**Response:** We deeply appreciate the reviewer's thoughtful comment. We have provided detailed information on the uncertainty calculation of input data, the selection criteria for the optimal solution and diagnostic plots. Besides, we have conducted the error estimates for the selected PMF solution, and found some uncertainties involved in this solution. To improve the reliability of the selected solution, we have added some constraints on the source profiles based on the priori information of the sources. Details of the PMF analysis are as follow.

**1. Uncertainties of the input data**

According to the User Guide of PMF5.0 (Norris et al., 2014), the uncertainties of the target species can be calculated as follow:

$$Unc = 5/6 \times MDL \ (c \le MDL) \tag{1}$$

$$Unc = \sqrt{(P \times c)^2 + (0.5 \times MDL)^2}$$
(2)

where *Unc* is the data uncertainty, c is the concentration of target species, MDL is the method detection limit, and P is the error fraction. Since the User Guide did not give the calculation method for the error fraction (P), we estimated the P values referring to

the measured relative standard deviations (RSD) of the target species. The RSD values were calculated by measuring six identical portions of an ambient sample. P was set as 10 % when RSD < 10 %, and 15 % or 20 % when RSD > 10 %.

**2. Selection of base solutions**

The chemical components input into the PMF model were selected based on our understanding of the possible WSOC sources (Norris et al., 2014). Interpretability was usually considered to be the most important factor for selecting the optimum solution (Shrivastava et al., 2007; Huang et al., 2014). Interpretable solutions are those which group tracers from different sources into distinct factors, while those solutions grouping tracers from multiple sources into one factor, distributing tracers for one source across multiple factors, or including factors with no distinct grouping of species are judged less interpretable (Shrivastava et al., 2007; Sowlat et al., 2016). In some literature, the optimal solution was defined as that with the maximum number of factors which had distinctive groupings of species, and explained at least 90 % of the total variable (Shrivastava et al., 2007).

In this study, PMF was run repeatedly by changing the number of factors and the start seed numbers. The base solution was selected based on: (1) the interpretability of the derived factor profiles and the temporal variations of source contributions; (2) the reconstruction of the total variable and R2 of input organic tracers (R2>0.90); (3) the scaled residuals of the input species. As presented in Figure S1, the 7-factor solution separated cholesterol (the tracer for cooking) into multiple sources. It was difficult to explain why cholesterol appeared in the factor profiles of biomass burning, dust and fresh biogenic SOC. Besides, this solution led to poor fits for cholesterol (R2 = 0.28) and *cis*-pinonic acid (R2 = 0.32), which were the key tracers selected in this study. Therefore, the 7-factor solution was not selected. As shown in Figure S2, the 8-factor solution also distributed cholesterol into multiple factors. This solution also resulted in a poor fit (R2 = 0.28) for cholesterol. Therefore, the 8-factor solution was not chosen in this study. As shown in Figures S5-7, the solutions with 4 to 6 factors all showed poor interpretability for the derived factor profiles and poor fits for the key

organic tracers. The 10-factor solution involved a factor without any tracer of high loading to indicate a specific source, thus could not be explained. By comparing the results with different factor numbers, the solution with 9 factors (Figure S3) was thought to be the most interpretable one.

Figure S1. A 7-factor solution resolved by the PMF model.

---

## Author Response (AR1)

**Response to Reviewer 1**

Overall comment: Yu et al reports observations of organic aerosol, both primary and secondary, collected on filters for different seasons/time periods of 2017 in Beijing, China. They report water soluble organic carbon (WSOC), its hydrophobic and hydrophilic portions, water soluble ions, total PM2.5, total organic carbon (OC), and total elemental carbon (EC). Further, they report tracers associated to different sources (levoglucosan, cholesterol, phthalic acid, 4-methyl-5-nitrocatechol, 2-methylerythritol, 3-hydroxyglutaric acid, and *cis*-pinonic acid). They use the tracers to differentiate sources of OC and WSOC during the seasons via "CO-scaled" concentrations, day and night ratios, correlation coefficients with various meteorological and chemical properties of aerosol ("acidity" and liquid water content), and positive matrix factorization. They find that aqueous chemistry explains a large portion of the secondary organic carbon during most seasons except summer, where photochemistry explains an important biogenic portion. They also find differences in the sources between the seasons (biomass burning vs dust vs vegetation). Overall, the paper is important and of interest to Atmsopheric Chemistry and Physics community as there is general overall uncertainty in the sources of organic aerosol in urban environments, especially during all seasons and high pollution events. The paper will be of value once the authors address the comments below.

**Response:** We deeply appreciate the reviewer's time and efforts devoted to improving this manuscript. We have provided our responses point by point below each comment, and have carefully revised the paper according to the reviewer's valuable suggestions.

1. Section 2.1 Sampling: Further information is needed here for the readers to have a better understanding of how the aerosol was collected–Was there a drier in-line prior to be collected on the filters? Was there a denuder to scrub gases prior to the filter to minimize gas-particle partitioning? Was there an impactor or cyclone for size selection? Further, of importance, was there any analysis of potential reactions that occurred on the filters prior to sampling?

**Response:** Thanks for the reviewer's comment. Our sampler included a  $PM_{2.5}$  impactor, but no in-line drier or denuder was used in this study.

Sampling of organic carbon is accompanied by both positive and negative artifacts. The positive artifact is due to the adsorption of gaseous organics to the sampling filter, and the negative artifact is caused by the evaporation of collected particulate organic carbon. To eliminate the positive artifact, a denuder can be placed upstream the sample filter to remove the gaseous organics by diffusion to the adsorbent surface (Cheng et al., 2009). The use of a denuder in the sampling system has been reported in previous studies (Eatough et al., 1993, 1999; Mader et al., 2001; Matsumoto et al., 2003; Viana et al., 2006; Cheng et al., 2009, 2010, 2012; Kristensen et al., 2016). The use of a denuder may induce a larger negative artifact, however, as the removal of gaseous organics can enhance the evaporation of particulate OC. Thus a backup filter should also be included in the sampling system (Cheng et al., 2009). Besides, the flow rate passing through the denuder was very low in most studies (Matsumoto et al., 2003; Viana et al., 2006; Cheng et al., 2009, 2010, 2012; Kristensen et al., 2016). This might be due to the significantly decreased removal efficiency of the denuder as the air flow rate increased (Cui et al., 1998; Ding et al., 2002). To collect enough samples for the accurate measurement of trace organic species, the flow rate of 1.05 m3 min-1 was chosen in this study. The air flow rate of about 1.05 m3 min-1 has been frequently used in the field sampling of organic aerosols (Kawamura et al., 2013; Verma et al., 2012, 2015; Li et al., 2018; Ma et al., 2018; Huang et al., 2020). At this flow rate, a denuder with a high removal efficiency is hardly commercially available.

Nevertheless, we were aware of the potential sampling artifacts and attempted to estimate the sampling artifacts of OC based on the literature results. Firstly, the adsorption behavior of OC might vary with meteorological conditions. Besides, the OC fractions with different volatility show different adsorption behavior. Cheng et al. (2015) compared the concentrations of different OC fractions (OC1, OC2, OC3, OC4) on bare quartz filters with those on denuded quartz filters in the four seasons of Beijing, and the results are summarized in Table S1. The contributions of different OC

fractions measured in this study are also listed in Table S1.

**Table S1** The ratio of the OC concentrations on the bare quartz filters to those on the denuded quartz filters in Cheng et al. (2015), as well as the contribution of different OC fractions measured in this study.

|        | The ra  | tio of OC o   | n bare quai | tz filters to | The con | The contribution of different OC fractions measured in this study |        |        |  |  |  |
|--------|---------|---------------|-------------|---------------|---------|-------------------------------------------------------------------|--------|--------|--|--|--|
|        | denuded | d quartz filt | ers (Cheng  | et al., 2015) | )       |                                                                   |        |        |  |  |  |
|        | OC1     | OC2           | OC3         | OC4           | OC1     | OC2                                                               | OC3    | OC4    |  |  |  |
| Winter | 1.27    | 1.03          | 1.02        | 1.05          | 10.8 %  | 19.6 %                                                            | 24.7 % | 44.8 % |  |  |  |
| Spring | 2.05    | 1.05          | 1.00        | 1.01          | 3.9 %   | 27.2 %                                                            | 43.1 % | 25.7 % |  |  |  |
| Summer | 2.45    | 1.60          | 1.17        | 1.08          | 4.4 %   | 37.6 %                                                            | 36.0 % | 22.0 % |  |  |  |
| Autumn | 2.08    | 1.05          | 0.99        | 1.01          | 7.9 %   | 26.5 %                                                            | 40.2 % | 25.3 % |  |  |  |

McDow (1986) systematically investigated the effect of sampling procedure on the OC measurement. The adsorption of organic vapors on bare quartz filters ( $C_{postive}$ artifact) was a function of the sampling duration (t) multiplied by the face velocity (v) as follows:

$$C_{postive artifact} = \sum_{i} \rho_{i} \frac{1 - e^{-\varepsilon_{i} v t}}{\varepsilon_{i} v t}$$
(1)

where the face velocity (v, cm s-1) is the ratio of the flow rate (cm3 s-1) to the sampling area of the filter (cm2),  $\rho_i$  is the concentration of adsorptive vapor *i* (g cm-3), and  $\varepsilon_i$  is a constant which can be defined as:

$$\varepsilon_i = \frac{1}{l} \left[ \beta f(t) t_o \exp\left(\frac{Q_A}{RT}\right) + 1 \right]$$
(2)

where l is the effective filter thickness. The average thickness of the quartz filter used in this study was 463  $\mu$ m. The other parameters are all constants.

Hence, it can be calculated that  $\varepsilon_i > 1/l > 20 \text{ cm}^{-1}$ , and  $1-e^{-\varepsilon vt} \approx 1$ . Therefore, the positive artifact (Cpositive artifact) is inversely proportional to the product of the sampling duration and the face velocity (v×t). The face velocity of Cheng et al. (2015) was 9.8 cm s-1, while that in our study was 47.3 cm s-1. The sampling duration of Cheng et al. (2015) was 24 h, while that in our study was 12 h. That is to say, the positive artifact of Cheng et al. (2015) was about 2.4 times higher than that in our study.

Based on the literature results and taking into account all the factors (seasons,

OC fractions, sampling procedure), the contribution of positive artifact to the measured OC was estimated to be 2.3 %, 1.4 %, 9.9 %, and 2.2 % in winter, spring, summer and autumn respectively in this study, which is roughly acceptable.

To further estimate the impact of gas-particle partitioning and potential reactions occurring on the filters, we overlapped two quartz filters and took samples at a flow rate of  $1.05 \text{ m}^3 \text{ min}^{-1}$  for a duration of 12 h. The organic tracers selected in this study were measured in both filters. The organic tracers on the backup filters typically originate from three sources: (1) adsorption of the organic vapors in the atmosphere; (2) adsorption of the semi-volatile species evaporated from the front filter; (3) secondary formation from the adsorbed organic vapors on the backup filter. Except for *cis*-pinonic acid, the tracer concentrations on the backup filter were all less than 5 % of those on the front filters, while the concentration of *cis*-pinonic acid on the backup filter was 21.6 % of that on the front filter. This result suggested that the sampling procedure in this study might bring some uncertainties for the measurement of *cis*-pinonic acid, and the sampling artifact was not significant for the other organic tracers.

**References**

- Cheng, Y., He, K., Duan, F., Zheng, M., Ma, Y., and Tan, J.: Positive sampling artifact of carbonaceous aerosols and its influence on the thermal-optical split of OC/EC, Atmos. Chem. Phys., 9, 7243-7256, 2009.
- Cheng, Y., He, K., Duan, F., Zheng, M., Ma, Y., Tan, J., and Du, Z.: Improved measurement of carbonaceous aerosol: evaluation of the sampling artifacts and inter-comparison of the thermal-optical analysis methods, Atmos. Chem. Phys., 10, 8533-8548, doi:10.5194/acp-10-8533-2010, 2010.
- Cheng, Y., Duan, F., He, K., Du, Z., Zheng, M., and Ma, Y.: Sampling artifacts of organic and inorganic aerosol: Implications for the speciation measurement of particulate matter, Atmos. Environ., 55, 229-233, doi:10.1016/j.atmosenv.2012.03.032, 2012.
- Cheng, Y., and He, K.: Uncertainties in observational data on organic aerosol: An annual perspective of sampling artifacts in Beijing, China, Environ. Pollut., 206, 113-121, doi:10.1016/j.envpol.2015.06.012, 2015.
- Cui, W., Eatough, D. J., and Eatough, N. L.: Fine particulate organic material in the Los Angeles Basin-I: Assessment of the high-volume Brigham Young University Organic Sampling System, BIG BOSS, J. Air Waste Manage Assoc., 48, 1024-1037, doi:10.1080/10473289.1998.10463760, 1998.

- Ding, Y., Pang, Y., and Eatough, D. J.: High-volume diffusion denuder sampler for the routine monitoring of fine particulate matter: I. design and optimization of the PC-BOSS, Aerosol Sci. Tech., 36, 369–382, 2002.
- Eatough, D. J., Wadsworth, A., Eatough, D. A., Crawford, J. W., Hansen, L. D., and Lewis, E. A.: A multiple-system, multichannel diffusion denuder sampler for the determination of fine particulate organic material in the atmosphere, Atmos. Environ., 27A, 1213–1219, 1993.
- Eatough, D. J., Obeidi, F., Pang, Y., Ding, Y., Eatough, N. L., and Wilson, W. E.: Integrated and real-time diffusion denuder sampler for PM2.5, Atmos. Environ., 33, 2835–2844, 1999.
- Kawamura, K., Tachibana, E., Okuzawa, E., Aggarwal, S. G., Kanaya, Y., and Wang,
  Z.: High abundances of water-soluble dicarboxylic acids, ketocarboxylic acids and α-dicarbonyls in the mountaintop aerosols over the North China Plain during wheat burning season, Atmos. Chem. Phys., 13, 8285-8302, doi:10.5194/acp-13-8285-2013, 2013.
- Kristensen, K., Bilde, M., Aalto, P. P., Petaja, T., and Glasius, M.: Denuder/filter sampling of organic acids and organosulfates at urban and boreal forest sites: Gas/particle distribution and possible sampling artifacts, Atmos. Environ., 130, 36-53, doi:10.1016/j.atmosenv.2015.10.046, 2016.
- Mader, B. T., Flagan, R. C., and Seinfeld, J. H.: Sampling atmospheric carbonaceous aerosols using a particle trap impactor/denuder sampler, Environ. Sci. Technol., 35, 4857-4867, 2001.
- Matsumoto, K., Hayano, T., and Uematsu, M: Positive artifact in the measurement of particulate carbonaceous substances using an ambient carbon particulate monitor, Atmos. Environ., 37, 4713-4717, doi:10.1016/j.atmosenv.2003.07.006, 2003.
- McDow, S. R.: The effect of sampling procedures on organic aerosol measurement, Ph.D. dissertation, Oregon Graduate Center, 1987.
- Viana, M., Chi, X., Maenhaut, W., Cafmeyer, J., Querol, X., Alastuey, A., Mikuska, P., and Vecera, Z.: Influence of sampling artifacts on measured PM, OC, and EC levels in carbonaceous aerosols in an urban area, Aerosol Sci. Tech., 40, 107-117, 2006.

2. Line 121: It is unclear what the standards curves were of (the tracers reported throughout paper or other standards), and what is meant by "standard curves with five to seven concentration gradients were re-established." What was re-established?

**Response:** Thanks for the reviewer's comment. The standard curves were made by the silylation derivatives of the organic tracers, and the detailed information of these standards is listed in Table S2. "Re-established" means that we measured the derivative products of the standard solutions each time before measuring the ambient samples. We have avoided the unclear expression and revised as follows:

"The authentic standards (Table S2) were dissolved in anhydrous pyridine, and diluted to five to seven different concentrations. Then 100  $\mu$ L of the standard solutions were reacted with 200  $\mu$ L silylating reagent (BSTFA: TMCS = 99:1) at 75°C for a duration of 70 min. After cooling down to the ambient temperature, these solutions containing derivative products were diluted to 1 mL with n-hexane, and measured by GC/MS/MS right before the analysis of ambient samples. The R2 of the derivative products were above 0.99, indicating good linearities of these standard curves. "

**3. Line 122: What authentic standards? What company/purity?**

**Response:** The information of the authentic standards is provided in Table S2 below.**Table S2** The detailed information of the authentic standards used in this study.

| Authentic standard       | Molecular formula                            | CAS number | Company                    | Purity |
|--------------------------|----------------------------------------------|------------|----------------------------|--------|
| Levoglucosan             | $C_{6}H_{10}O_{5}$                           | 498-07-7   | Sigma-Aldrich              | 99 %   |
| Cholesterol              | C 27 H 46 O            | 57-88-5    | Sigma-Aldrich              | 93 %   |
| Phthalic acid            | $C_8H_6O_4$                                  | 88-99-3    | Sigma-Aldrich              | 99 %   |
| 4-Methyl-5-nitrocatechol | C7H7NO4                                      | 68906-21-8 | Toronto Research Chemicals | 98 %   |
| 2-Methylerythritol       | C5H12O4                                      | 58698-37-6 | Sigma-Aldrich              | 90 %   |
| 3-Hydroxyglutaric acid   | C 5 H 8 O 5 | 638-18-6   | Sigma-Aldrich              | 95 %   |
| cis-Pinonic acid         | $C_{10}H_{16}O_3$                            | 61826-55-9 | Sigma-Aldrich              | 98 %   |

4. Section 2.3: Please describe or cite the PMF software used. PMF 5.0 is not enough to understand how positive matrix factorization was actually conducted.

**Response:** Thanks for this kind suggestion. We have added a brief introduction to the PMF software in the manuscript. Besides, we have added the detailed information on the uncertainty calculation of the input data, the selection criteria for the optimal solution, the diagnostic plots and error estimation in the supplementary material.

5. Line 140: I highly recommend the use of "aerosol acidity," as defined in this line, due to the discussion from Pye et al. 2020 (https://acp.copernicus.org/articles/20/4809/2020/). The ratio here does not define acidity, and is analytically challenging to say if it is defining the amount of hydronium ions in the aerosol phase, as the hydronium ions may be a very low detection limits that cannot be quantified due to propagation of uncertainty.

**Response:** We deeply appreciate the reviewer's valuable suggestion. According to Pye et al. (2020), we changed the ratio of  $R_{A/C}$  to the approximate value of aerosol pH (pHF) to denote the aerosol acidity. The pHF value was estimated using the ISORROPIA-II model. And the molality of H+ (mH+), which was calculated by mH+ = 10-pHF, was used instead for the correlation analysis in the revised manuscript.

6. Section 3: Though an important and valuable aspect of this whole manuscript is that the filters were collected during different seasons, I highly recommend the authors soften the language throughout that the results "reflect" a specific season or are similar or different to other studies. Since it's only for one year and approximately 2 weeks for each season. The limited data makes it hard to say how typical the results are and this should be discussed/emphasized throughout (instead of general statements that in fall this is what is observed/happens).

**Response:** Thanks for the reviewer's valuable suggestion. We have changed the terms of "winter/spring/summer/autumn" to "January/April/July/October" or "during the sampling periods in winter/spring/summer/autumn" throughout the revised manuscript, in order to be more specific.

7. Another area I suggest the authors be careful in their discussion of r values, as majority of the values they report lead to R2 values less than 0.5 (thus explaining less than 50% of the variability observed).

**Response:** We agree with the reviewer that most of the r values led to  $R^2$  less than 0.5, thus could only explain less than 50 % of the variability observed, and were insufficient to reach clear conclusions. Therefore, in the revised manuscript, we only explained the r results and avoided making definite conclusions based on the r values.

8. Line 195: Since the authors are comparing OC from emissions inventory to Fig. 2, I would recommend converting the emissions to OC-to-CO ratios. Also, I would

recommend adding these ratios, if possible, to Fig. 2, for direct comparisons with observations.

**Response:** We are deeply grateful for the reviewer's valuable suggestion. As suggested by the reviewer, we have added the emissions of OC from the Fire Inventory to Figure 2 in the revised manuscript. As a tracer for biomass burning, levoglucosan in the atmosphere may derive from both residential biofuel burning and open biomass burning. To determine whether open biomass burning was the dominant type of biomass burning in Beijing, the seasonal variation of the CO-scaled concentration of levoglucosan was compared with that of the OC emission amount from the Fire Inventory in this study. The CO-scaled concentration of levoglucosan showed a totally different seasonal variation trend from that of the OC emission from open biomass burning, therefore, we speculated that open biomass burning was not the dominant category of biomass burning in Beijing. After careful consideration, we thought that it was not necessary to convert the emissions to OC-to-CO ratios. Instead, the absolute emission amounts of OC from the Fire Inventory were added.

**Figure 2.** (a) The CO-scaled concentration of the identified organic tracers; (b) The day to night ratios of the measured concentrations of the organic tracers; (c) The OC emission amounts from open biomass burning provided by the Fire Inventory (FINN)

in Beijing during the sampling periods in four seasons of 2017.

9. Line 199: Is it possible to get emission inventory values of residential biofuel combustion and coal combustion to compare with the OC from open biomass burning?

**Response:** The total emission amounts of OC from the residential sources in January, April, July and October of 2016 can be obtained from the multi-resolution emission inventory for China (MEIC, http://www.MEICmodel.org). However, the residential OC from MEIC is a total amount of biofuel combustion, coal combustion, other fossil fuel combustion, etc. Unfortunately, the detailed OC emission amount from each residential source is not available. Therefore, the respective OC emission inventory values of residential biofuel combustion and coal combustion were not provided in the manuscript.

10. Line 207: It is unclear how aromatics form SOA to impact WSOC during winter, as the photochemistry is greatly reduced. Could the authors provide more discussion concerning this?

**Response:** We deeply appreciate the reviewer's valuable suggestion. The following discussion has been added in the revised manuscript.

Although the concentrations of oxidants were usually lower in winter due to the weaker solar radiation, a previous observation found that the  $\cdot$ OH concentration in Beijing was significantly higher than that in New York, Birmingham and Tokyo, and was nearly 1 order of magnitude higher than that predicted by global models in northern China in winter (Tan et al., 2018). Zhang et al. (2019) indicated that HONO, which was mainly from the heterogeneous reactions of NO2 and traffic emissions, was the major precursor of  $\cdot$ OH in winter. According to the WRF-Chem model simulation, HONO resulted in a significant enhancement (5-25 µg m-3) of SOA formation (most of which were from the aromatic precursors) during a haze episode in winter in the Beijing-Tianjin-Hebei region (Zhang et al., 2019). Besides, some recent studies suggested that the brown carbon-derived singlet molecular oxygen ( $^{1}O_{2}^{*}$ ) in aerosol

liquid water could react rapidly with the electron-rich organics such as PAHs, thus facilitate the aromatic SOA formation (Kaur et al., 2019; Manfrin et al., 2019). This process might be more significant in winter, when the concentration of HULIS was much higher than that in other seasons. Therefore, both the enhanced levels of oxidants including  $\cdot$ OH and  ${}^{1}O_{2}{}^{*}$  and the higher concentrations of aromatic precursors in winter contributed to the enhanced aromatic SOA formation during the study period in winter.

**References**

- Tan, Z., Rohrer, F., Lu, K., Ma, X., Bohn, B., Broch, S., Dong, H., Fuchs, H., Gkatzelis, G. I., Hofzumahaus, A., Holland, F., Li, X., Liu, Y., Liu, Y., Novelli, A., Shao, M., Wang, H., Wu, Y., Zeng, L., Hu, M., Kiendler-Scharr, A., Wahner, A., and Zhang, Y.: Wintertime photochemistry in Beijing: observations of ROx radical concentrations in the North China Plain during the BEST-ONE campaign, Atmos. Chem. Phys., 18, 12391-12411, doi:10.5194/acp-18-12391-2018, 2018.
- Zhang, J., Chen, J., Xue, C., Chen, H., Zhang, Q., Liu, X., Mu, Y., Guo, Y., Wang, D., Chen, Y., Li, J., Qu, Y., and An, J.: Impacts of six potential HONO sources on HOx budgets and SOA formation during a wintertime heavy haze period in the North China Plain, Sci. Total Environ., 681, 110-123, doi:10.1016/j.scitotenv.2019.05.100, 2019.

11. Line 249: It is surprising that the authors are saying that gas-phase photooxidation was not the dominant formation mechanism of secondary organic carbon. I can see maybe WSOC, but seeing all secondary organic carbon is a big statement. Especially, since the authors go on in line 254-55 to say photochemistry plays a role.

**Response:** We deeply appreciate the reviewer's valuable comment. We agree with the reviewer that it was not proper to conclude that "gas-phase photooxidation was not the dominant formation mechanism of SOC" merely based on the result that "WSOC/OC did not show any significant positive correlation with O3 concentrations". The corresponding statement has been deleted in the revised manuscript. Besides, "SOC" has been corrected to "SOC in WSOC" to be more specific.

12. Line 339: Source 3 did not show the highest contribution in winter....highest contribution of what?

**Response:** We feel sorry for the unclear expression. This sentence has been revised as: "The primary emission strength of coal combustion was the strongest in winter among four seasons, since the domestic heating activities required extra amounts of coal combustion in this season. However, the contribution of Factor 3 to WSOC during the study period in winter was not the highest among four seasons, implying that there could be other sources beyond coal combustion included in Factor 3.

13. Line 416-419: I would recommend caution here, as other hypothesis have been stated for reasons in differences between chamber SOA and ambient SOA, including losses of vapors to the walls and autoxidation (which has been shut down in chamber experiments due to too high NOx levels and/or too high aerosol loadings).

**Response:** We deeply appreciate the reviewer's valuable suggestion. This statement has been revised as: "Previous observation suggested that a large fraction of ambient SOA was more oxidized than those generated in the dry smog chambers, where SOA could only be produced through gas-phase oxidation (Aiken et al., 2008). There have been some hypothesis for the difference between the chamber SOA and ambient SOA, such as the losses of vapors to the walls, and the autoxidation due to the uncertainties in chamber radical environment (McVay et al., 2016; Thornto et al., 2020). Besides, the results of this study also indicated that the aqueous-phase processing, which can produce more hydrophilic SOA, may be one of the reasons for the discrepancy in the oxidation degrees of ambient SOA and chamber SOA (Ervens et al., 2011)."

**References**

- McVay, R.C., Zhang, X., Aumont, B., Valorso, R., Camredon, M., La, Y. S., Wennberg, P. O., and Seinfeld, J. H.: SOA formation from the photooxidation of α-pinene: systematic exploration of the simulation of chamber data, Atmos. Chem. Phys., 16, 2785-2802, doi:10.5194/acp-16-2785-2016, 2016.
- Thornton, J. A., Shilling, J. E., Shrivastava, M., D'Ambro, E. L., Zawadowicz, M. A., and Liu, J.: A near-explicit mechanistic evaluation of isoprene photochemical secondary organic aerosol formation and evolution: simulations of multiple chamber experiments with and without added NOx, ACS Earth Space Chem., 4, 1161-1181, doi: 10.1021/acsearthspacechem.0c00118, 2020.
- 14. Table 1: I would recommend somehow highlighting which values show statistical

differences between day and night and between seasons. Also, I would recommend including average CO mixing ratios.

**Response:** We deeply appreciate the reviewer's suggestion. The average values which showed statistical differences among seasons and between day and night have been highlighted as follows, and the average CO concentrations have also been included in Table 1 in the revised manuscript.

| Compounds (ug m -3 )       | Winter          |                 |                              |                     | Spring             |                            |                     | Summer              |                          |                    | Autumn             |                          |
|---------------------------------------|-----------------|-----------------|------------------------------|---------------------|--------------------|----------------------------|---------------------|---------------------|--------------------------|--------------------|--------------------|--------------------------|
|                                       | Daytime         | Nighttime       | Mean                         | Daytime             | Nighttime          | Mean                       | Daytime             | Nighttime           | Mean                     | Daytime            | Nighttime          | Mean                     |
| СО                                    | $1.7 \pm 1.7$   | $2.2 \pm 2.8$   | $1.9\pm2.3^{\rm a}$          | $0.6\pm0.4^{*}$     | $0.8\pm0.4^{\ast}$ | $0.7\pm0.4^{\rm b}$        | $1.1\pm0.4$         | $0.9\pm0.2$         | $1.0\pm0.3^{\text{b}}$   | $1.1\pm0.4^{*}$    | $1.4\pm0.4^{\ast}$ | $1.2\pm0.4^{\text{b}}$   |
| PM 2.5                     | $120\pm107$     | $147\pm154$     | $133\pm131^{\rm a}$          | $60.6 \pm 36.2$     | $64.5\pm34.8$      | $62.5\pm34.9^{b}$          | $59.8\pm28.6$       | $51.9\pm20.6$       | $55.8\pm24.8^{b}$        | $75.2\pm58.1$      | 81.1 ± 50.8        | $78.2\pm53.7^{\text{b}}$ |
| OC                                    | $20.1 \pm 19.2$ | $21.0 \pm 24.8$ | $20.6\pm21.9^{\rm a}$        | $7.9\pm2.6$         | $9.5\pm3.4$        | $8.7\pm3.1^{\text{b}}$     | $8.7\pm3.4^{*}$     | $6.8\pm4.3^{\ast}$  | $7.8\pm3.9^{b}$          | $9.4 \pm 3.8$      | $10.1 \pm 3.7$     | $9.7\pm3.7^{b}$          |
| EC                                    | $3.9 \pm 3.1$   | $4.7\pm5.8$     | $4.3\pm4.6^{\rm a}$          | $1.9 \pm 1.1^*$     | $2.7\pm\!\!1.4^*$  | $2.3\pm1.3^{\text{b, c}}$  | $1.4 \pm 1.0$       | $1.3 \pm 1.0$       | $1.3 \pm 1.0^{\circ}$    | $2.4\pm1.4^*$      | $3.4\pm1.7^*$      | $2.9\pm1.6^{\text{b}}$   |
| OC/EC                                 | $4.6 \pm 1.1$   | $4.3 \pm 1.3$   | $4.5\pm1.2^{a,b}$            | $5.2 \pm 2.1$       | $4.4\pm2.5$        | $4.8\pm2.3^{a,b}$          | $6.7\pm3.9$         | $5.3 \pm 4.2$       | $6.1\pm4.1^{a}$          | $4.4 \pm 1.6^{*}$  | $3.3\pm0.9^{\ast}$ | $3.8\pm1.4^{\text{b}}$   |
| WSOC                                  | $11.4 \pm 11.3$ | $12.0 \pm 16.4$ | $11.7 \pm 13.9^{\mathrm{a}}$ | $4.1\pm2.0$         | $4.7\pm2.6$        | $4.4\pm2.3^{b}$            | $5.3\pm2.1^{\ast}$  | $4.0\pm2.7^{\ast}$  | $4.7\pm2.5^{\rm b}$      | $4.7\pm3.0$        | $4.9\pm2.8$        | $4.8\pm2.8^{\text{b}}$   |
| WSOC/OC                               | $0.53\pm0.08$   | $0.51 \pm 0.08$ | $0.52\pm0.08^{\text{b}}$     | $0.50 \pm 0.10$     | $0.47\pm0.14$      | $0.49\pm0.12^{b}$          | $0.62\pm0.11$       | $0.59\pm0.10$       | $0.60\pm0.11^{a}$        | $0.47 \pm 0.12$    | $0.46\pm0.12$      | $0.46\pm0.12^{\text{b}}$ |
| MH-WSOC                               | $7.9\pm7.6$     | $8.0\pm10.3$    | $8.0\pm8.9^{\rm a}$          | $2.8\pm1.3$         | $2.9\pm1.6$        | $2.9\pm1.5^{\text{b}}$     | 4.1 ± 1.2           | $3.4 \pm 1.6$       | $3.8\pm1.5^{\text{b}}$   | $2.9\pm1.6$        | $2.9 \pm 1.3$      | $2.9\pm1.5^{\text{b}}$   |
| SH-WSOC                               | $3.2\pm3.8$     | $4.0\pm 6.1$    | $3.6\pm5.0^{\rm a}$          | $1.3\pm0.9$         | $1.8 \pm 1.1$      | $1.6 \pm 1.0^{\rm b}$      | $1.2\pm1.0^{*}$     | $0.7 \pm 1.1^{*}$   | $1.0 \pm 1.1^{\text{b}}$ | $1.8 \pm 1.4$      | $2.0 \pm 1.5$      | $1.9\pm1.4^{\text{b}}$   |
| Organic tracers (ng m -3 ) |                 |                 |                              |                     |                    |                            |                     |                     |                          |                    |                    |                          |
| Levoglucosan                          | $307\pm300$     | $388\pm394$     | $349\pm348^{a}$              | $100 \pm 87.8^{*}$  | $194 \pm 175^{*}$  | $147\pm144^{b}$            | $23.6\pm11.0$       | $34.2\pm24.2$       | $28.9 \pm 19.3^{\circ}$  | $136\pm102^*$      | $234\pm125^*$      | $185 \pm 123^{b}$        |
| Cholesterol                           | $5.0 \pm 3.0$   | $4.9\pm3.3$     | $4.9\pm3.1^{a,b}$            | $3.9\pm1.9$         | $4.8\pm2.5$        | $4.3\pm2.3^{b}$            | $4.1 \pm 2.4$       | $3.0 \pm 1.1$       | $3.6\pm1.9^{\text{b}}$   | $6.1 \pm 4.4$      | $6.3 \pm 3.1$      | $6.2\pm3.8^{\rm a}$      |
| Phthalic acid                         | $88.7\pm84.8$   | $90.8 \pm 121$  | $89.8\pm103^{\rm a}$         | $27.3\pm20.8$       | $21.9 \pm 14.0$    | $24.6\pm17.7^{\mathrm{b}}$ | $55.9\pm22.0^*$     | $17.6\pm9.1^{\ast}$ | $36.8\pm25.5^{\text{b}}$ | $27.6 \pm 21.8$    | 19.9 ± 13.3        | $23.8\pm18.2^{\text{b}}$ |
| 4-Methyl-5-nitrocatechol              | $24.7\pm26.4$   | $35.2 \pm 41.0$ | $30.1\pm34.5^{\mathrm{a}}$   | $1.8\pm1.9$         | $3.3\pm2.7$        | $2.6\pm2.4^{\text{b}}$     | $0.1\pm0.3$         | $0.0\pm0.0$         | $0.1\pm0.2^{\text{b}}$   | $1.6 \pm 1.2^{*}$  | $4.4\pm3.6^{*}$    | $3.0\pm3.0^{\text{b}}$   |
| 2-Methylerythritol                    | $2.1 \pm 2.3$   | $2.2 \pm 3.5$   | $2.2\pm2.9^{\rm b}$          | $1.2\pm0.6$         | $1.5\pm0.8$        | $1.4\pm0.7^{b}$            | $55.4\pm48.5$       | $41.6 \pm 34.6$     | $48.5\pm42.0^{\rm a}$    | $2.3 \pm 1.1$      | $2.6 \pm 1.2$      | $2.5\pm1.1^{\text{b}}$   |
| 3-Hydroxyglutaric acid                | $4.4\pm3.9$     | $4.2 \pm 5.0$   | $4.3\pm4.5^{\rm b}$          | $4.2\pm2.8$         | $4.9\pm5.1$        | $4.6\pm4.0^{b}$            | $37.1 \pm 22.7^{*}$ | $27.3 \pm 18.5^{*}$ | $32.2\pm20.9^{\rm a}$    | $7.5 \pm 4.6$      | $7.0 \pm 4.4$      | $7.2\pm4.5^{b}$          |
| cis-Pinonic acid                      | $3.3\pm2.4$     | $3.0 \pm 2.1$   | $3.2\pm2.2^{\circ}$          | $9.0\pm 6.0^{\ast}$ | $6.9\pm3.6^*$      | $7.9\pm5.0^{\rm a}$        | $7.3 \pm 4.2$       | $10.1\pm6.0$        | $8.7\pm5.3^{\rm a}$      | $7.3\pm3.0^{\ast}$ | $3.6\pm0.8^{\ast}$ | $5.5\pm2.9^{b}$          |

Table 1 The average concentrations and standard deviations of the identified carbonaceous species in PM2.5 during the sampling periods in four seasons.

a, b, c We performed one-way analysis of variance (ANOVA) to evaluate whether these mean values showed statistically significant differences (p<0.05) between two seasons. If two seasonal average values have one or more same superscripts, it means that they did not show significant differences (p>0.05). In contrast, if two average values do not have any same superscript, it means that they showed significant differences (p<0.05). For example, the PM2.5 concentration was significantly higher in winter than in other seasons, but it did not show significant difference between spring and autumn, or summer and autumn. Besides, the EC concentration did not show significant difference between spring and autumn, but it showed significant difference between summer and autumn.

\* We also performed paired t test to evaluate whether daytime and nighttime values showed statistically significant differences (p

---

## Author Response (AR2)

**Response to referees**

We would like to express our sincere appreciation again for the referees' time and efforts devoted to improving this manuscript. The minor errors pointed out by referee 2 has been revised as follows:

**Comment 1**:L60, the meaning of "massive data analysis" is vague. Revise.

**Response:** We deeply appreciate this valuable suggestion. This sentence has been revised as follows: "The AMS-PMF method is based on the PMF analysis for the mass spectral of organic aerosols, which is usually a big dataset since online AMS has a high time-resolution".

**Comment 2**:L486, should be "Thornton et al., 2020"

**Response:** We deeply appreciate this kind suggestion. This error has been corrected.